# High-spatial resolution probability maps of drought duration and magnitude across Spain

Fernando Domínguez-Castro[1], Sergio M. Vicente-Serrano[1], Miquel Tomás-Burguera[2], Marina Peña-Gallardo[1], Santiago Beguería[2], Ahmed El Kenawy[1,3,4], Yolanda Luna[5], Ana Morata[5]

[1]Instituto Pirenaico de Ecología, Spanish National Research Council (IPE-CSIC), Zaragoza, 50059, Spain
[2]Estación Experimental de Aula Dei, Spanish National Research Council (EEAD-CSIC), Zaragoza, 50059, Spain
[3]Department of Geography, Mansoura University, Mansoura, 35516, Egypt
[4]Department of Geography, Sultan Qaboos University, Alkhud, 12317 Oman.
[5]Agencia Estatal de Meteorología (AEMET), Madrid, 28071, Spain

*Correspondence to*: Fernando Domínguez-Castro (f.dominguez.castro@gmail.com)

**Abstract.** Assessing the probability of occurrence of drought is important for improving current drought assessment, management and mitigation measures and strategies across Spain. As such, we employed two well-established drought indices, the Standardized Precipitation Index (SPI) and the Standardized Precipitation Evapotranspiration Index (SPEI), to characterize drought duration and magnitude at different time scales over Spain. In order to map drought hazard probability, we applied the extreme value theory and tested different thresholds to generate the peak-over-threshold drought (POT) duration and magnitude series. Our results demonstrate that the Generalized Pareto (GP) distribution performs well in estimating the frequencies of drought magnitude and duration. In specific, we found a good agreement between the observed and modelled data when using upper percentiles to generate the POT series. Spatially, our estimations suggest a higher probability of extreme drought events in southern and central areas of Spain, compared to the northern and eastern regions. Also, our study found spatial differences in drought probability estimations, as a function of the selected drought index (i.e. SPI vs. SPEI) and timescale (i.e. 1-, 3-, 6- and 12-month). The obtained drought hazard probability maps for Spain can contribute to better management of different sectors (e.g. agriculture, water resources management, urban water supply, and tourism) at national, regional and even local scale.

## 1 Introduction

Drought is one of the main hydro-climatic hazards in Spain, with adverse impacts on natural and human environments (Pérez and Barreiro-Hurlé, 2009; UNEP, 2006). Numerous studies have analysed drought characteristics in Spain, suggesting a strong variability over both space and time (e.g. Domínguez-Castro et al., 2018; González-Hidalgo et al., 2018). In Spain, drought management measures are usually based on insurances and government subsidies to diminish their impacts, particularly those related to agricultural sector (Fernández, 2006). In addition to the different monitoring systems for hydrological drought conditions (Maia and Vicente-Serrano, 2017), there are national legislation practices that aim to improve drought adaptation strategies and practices, such as Special Droughts Plans (Garrick et al., 2017).

Although current national measures are quite useful to diminish drought risk, other improved approaches are still needed to reduce drought risk, particularly those focusing on real time drought monitoring (e.g. Svoboda et al., 2002) and forecasting (e.g. Mishra et al., 2009; Mishra and Singh, 2011). In this context, drought probabilities maps can be a promising tool to characterise drought risk at a detailed spatial scale. In particular, it is possible to determine the probability of drought episodes of certain severity, which allows for establishing better sectorial management strategies. Due to the availability of dense spatial climatic data, there is also a possibility to map drought probabilities at fine spatial scale, which can be useful for different socioeconomic sectors and for managing natural ecosystems. The utility of probabilistic approaches to enhance drought monitoring and adaptation has been already evidenced in many regions worldwide (e.g. Engeland et al., 2005; Hussain et al., 2018; She et al., 2014; Tosunoglu and Can, 2016; Zamani et al., 2015).

In Spain, several studies have developed dry spells probability maps (e.g. Lana et al., 2006; Martin-Vide and Gomez, 1999; Pérez-Sánchez and Senent-Aparicio, 2018). However, given that the probability of occurrence of dry spells is higher in arid regions than in humid regions, these studies did not account for the different drought hazard probability across Spain. It is well-recognized that the frequency and duration of dry spells are closely driven by the climatology of the studied area. As such, it can be expected that a simple map of climate aridity in Spain can show similar spatial patterns to those of dry spell probability. However, drought probability caannot necessarily be related to the spatial patterns of climate aridity, but it is rather associated with the intrinsic characteristics of drought events recorded in each region. This is simply because, irrespective of the climatology, drought can occur in any world region when there is a negative anomaly with respect to the long-term average moisture conditions (Wilhite and Pulwarty, 2017). This highlights the importance of data standardization to make drought characteristics (e.g. duration, intensity, severity) comparable among regions with different climatic conditions. Several drought indices (e.g. Standardized Precipitation Index, Standardized Precipitation Evapotranspiration Index, Palmer Drought Severity Index, Self-calibrated Palmer Drought Severity Index) have been developed to characterise drought conditions across regions with different climatic conditions (Redmond, 2002). Also, irrespective of climatic conditions, these indices can identify drought episodes according to their duration and magnitude (Dracup et al., 1980) Overall, based on these drought indices, it is possible to map the probability of occurrence of drought duration and magnitude at a detailed spatial resolution. In their assessment of drought characteristics in Serbia, Tošić and Unkašević (2014) analysed probability of occurrence of drought using the SPI between 1949 and 2011, concluding that the Generalized Pareto (GP) distribution fits well with the series at 1- and 12- month timescales. Similarly, Yusof et al. (2013) analysed the probability of drought duration and magnitude using the SPI and rainfall data from 30 rain gauges distributed across the peninsular Malaysia. Zin et al. (2013) also analysed the return period for drought severity over the peninsular Malaysia by means of the SPI. A quick inspection of these studies reveals that they employed an individual drought index (SPI in most cases), with few attempts to explore the possible differences in drought hazard probability, as a function of different drought indices (e.g. Yan et al., 2018) or different drought timescales (Moradi et al., 2011; Tošić and Unkašević, 2014). Due to the varying response of the different hydrological sub-systems, socioeconomic sectors and natural ecosystems to drought timescales, drought timescales should be taken into account when determining drought impacts (McKee et al.,

1993; Vicente-Serrano, 2013). Moreover, the spatial patterns of drought and correspondingly drought hazard probability maps can differ largely, as a function of timescale (Vicente-Serrano, 2006). Thus, it is important to assess drought hazard probability at different drought timescales to meet the specific needs of the different socioeconomic sectors and natural systems.

The overall objective of this study is to employ a newly developed high-resolution spatial (1.21 km$^2$) and temporal (weekly) gridded dataset of drought indices (Vicente-Serrano et al., 2017) to characterise drought events in Spain. Specifically, this study aims to i) apply the extreme value theory to determine the best threshold and statistical distribution to fit the probability of drought duration and magnitude, ii) explore spatial variations in this probability as a function of two common drought indices, with different underlying calculations (i.e. SPI vs. SPEI), and iii) assess whether there are spatial differences

in drought hazard probability in response to the different drought timescales. In Spain, such a detailed spatial assessment is still lacking, which limits the possibility to provide guidance on the use of drought hazard probability to manage and mitigate drought risks at the national, regional and even local scale.

## 2 Data and methods

### 2.1 Dataset

Based on gridded datasets of maximum and minimum air temperatures (1304 observatories), precipitation (2269 observatories), wind speed (82 observatories), relative humidity (179 observatories) and sunshine duration (112 observatories), Vicente-Serrano et al. (2017) developed a high-resolution spatial (1.21 km$^2$) and temporal (weekly) drought dataset for Spain (412178 pixels). This dataset spans the period from 1961 to 2014. Importantly, this drought dataset was developed after a rigorous procedure to check the quality and homogeneity of the input climatic data. The grid of each

variable was computed by universal kriging method (Borrough and McDonnell 1998; Pebesma, 2004), using latitude, longitude and elevation of each grid cell as auxiliary variables. The grid layers were validated with a jackknife resampling procedure (Phillips et al., 1992) and difference between the predicted and observed values for each observatory was low (see Vicente-Serrano et al., 2017b for details). Overall, the gridded climatic data were employed to compute the Standardized Precipitation Index (SPI) (McKee et al., 1993) and the Standardized Precipitation Evapotranspiration Index (SPEI) (Vicente-

Serrano et al., 2010) at different timescales ranging from 1- to 48-month (http://monitordesequia.csic.es). While the SPI accounts only for precipitation data, the SPEI is based on normalization of the difference between precipitation and atmospheric evaporative demand (AED). In this study, we employed these two drought indices to assess the possible impacts of the AED on drought hazard probability in Spain. The SPI and SPEI were used at time scales of 1-, 3-, 6- and 12-months for the period 1961-2014.

## 2.2 Selection of drought events

There are several criteria (thresholds) to identify independent drought events (e.g. Fleig et al., 2006; Lee et al., 1986). These thresholds are generally arbitrary, with no clear objective metrics to relate a certain value of a drought index with specific sectorial impacts. Indeed, this is a challenging task, given the large number of economic sectors and environmental systems impacted by droughts (Pérez and Barreiro-Hurlé, 2009). Furthermore, regions and sectors can respond differently to various drought timescales (Lorenzo-Lacruz et al., 2013; Pasho et al., 2012). In this work, we obtained the series of drought events from the weekly gridded series of SPEI and SPI at four selected time scales (1-, 3-, 6- and 12-months). We used zero threshold to define drought events. Although this threshold allows for inclusion of less severe drought events, it can secure a sufficient sampling size to conduct the probabilistic analysis. Importantly, the retention of drought events in this manner will not bias the obtained results, given that high values of the series will be retained following the peak-over-threshold approach. Overall, each drought event was defined as the period of consecutive weeks with SPI or SPEI values lower than zero. Likewise, the series of drought duration and magnitude were created based on the consecutive weeks of SPEI/SPI values below zero. The drought magnitude was calculated following the classical approach of Dracup et al. (1980). However, for operative purposes, the total magnitude of drought was transformed to positive values.

## 2.3 Probabilistic analysis

The series of peaks-over-threshold (POT) were obtained using the series of drought duration and magnitude calculated at 1-, 3-, 6- and 12-month timescales. These series are stationary and do not show any trend (Domínguez-Castro et al., 2018), which is a prerequisite to apply the extreme value theory. The POT series are obtained according to a threshold ($x_0$), as:

$$Y = X - x_0 \forall X > x_0 \tag{1}$$

In order to assess the role of the selected threshold in fitting the probability distribution of the series, we tested different thresholds selected using the percentiles of the series (i.e. 0th, 10th, 20th,…, 90th and 95th). Following this procedure, we defined the optimal percentile threshold to define the exceedance series of drought duration and magnitude for the two drought indices and the four selected timescales.

Numerous studies employed the GP distribution to model meteorological and hydrological droughts (e.g. Fleig et al., 2006; Nadarajah, 2008; Nadarajah and Kotz, 2008; Chen et al., 2011; Yusof et al., 2013; Tošić and Unkašević, 2014; Trenberth et al., 2014; Zamani et al., 2015; Liu et al., 2016). This is mainly because the probability distribution of a POT series, with random occurrence times, fits well with GP distribution (e.g. Hosking et al., 1987; Pham et al., 2014; Wang, 1991). The GP distribution is a flexible, long-tailed distribution, whose distribution function is formulated, as:

$$F(x) = 1 - \left[1 - \frac{\kappa}{\alpha}(x - \varepsilon)\right]^{1/\kappa} \tag{2}$$

where $\kappa$, $\alpha$ and $\varepsilon$ are the shape, scale and location parameters of the distribution origin that corresponds to the lower bound x0. The GP parameters are obtained using the L-moment statistics following Hosking (1990).

Hosking (1990) proposed a procedure to provide parametric approximations to the relationships between L-skewness and L-kurtosis. This procedure allows for determining the suitability of the GP distribution to fit the exceedance obtained from different x0 values. Herein, we plotted the different L-moment diagrams with the statistics obtained from drought duration and magnitude series. The aim was to assess the suitability of different x0 thresholds to obtain POT series with good fitting to GP distribution.

We applied the Anderson-Darling test to check the goodness of fitting of the POT series obtained from different x0 thresholds. To define the most suitable threshold, we paid much attention to secure a sample of sufficient length to obtain solutions for the GP parameters. This is important to obtain reliable probabilistic estimations. For this purpose, we compared the observed maximum drought duration and magnitude obtained for the whole study period (1961-2014) with those estimated using GP distribution and POT for the different thresholds. We calculated the probability that an event of magnitude $X_T$ in a period of T = 54 years (expressed in the original scale) will occur at least once in a period of T years. This is formulated, as:

$$X_T = \varepsilon + \frac{\alpha}{\kappa}\left[1 - \left(\frac{1}{\lambda T}\right)^{\kappa}\right] \tag{3}$$

where $\lambda$ is a frequency parameter equalling the average number of occurrences of X per year in the original sample. The performance of each threshold was assessed using different accuracy statistics, including the mean absolute error (MAE), the Willmot' D agreement index (Willmott, 1981), and the Pearson's r correlation coefficient comparing the maximum observed drought duration and magnitude with the GP estimations for the same sample length.

Once a general threshold was established to define the POT series of drought duration and magnitude , we determined the goodness of the GP modelling for each drought index and timescale. For this purpose, we used the probability-probability (p-p) plots, which define the extent to which the empirical and modelled GP cumulative distribution functions (cdfs) closely match. This procedure was applied to a total of 412,178 gridded series of drought magnitude and duration covering the four selected timescales of both the SPI and SPEI. The empirical cdfs were obtained using the plotting position formula proposed by Hosking (1990) for highly skewed data, as according to:

$$P(X \leq x) = \frac{i - 0.35}{N} \tag{4}$$

where i is the rank of the observations arranged in descending order, and N is the number of observations. The goodness of agreement between the empirical and modelled cdfs was determined by means of a weighted correlation coefficient. This procedure gives more weight to the highest and less-frequent observations in the sample, which are more relevant to extreme values analysis. The weight was defined using the empirical cdf, as:

$$w_j = \frac{1}{1 - cdf(j)} \tag{5}$$

where cdf is the cumulative distribution function, and j are the observations in the series of exceedances sorted in ascending order.

# 3 Results

## 3.1 Selection of the distribution and threshold to define the POT series

Figure 1 illustratess some examples of L-moment diagrams considering the 1-month SPEI duration series over the peninsular Spain. The series for each diagram were obtained considering POT at different percentiles. Each line represents a theoretical curve distribution: the generalized logistic (GLO, blue), generalized extreme value (GEV, green), generalized Pareto (GPA, red), generalized normal (GNO, black) and Pearson type III (PE3, light blue). As noted, irrespective of the selected threshold, the drought duration series tend to closely approximate to GP distribution. Notably, there is a higher dispersion of points around the theoretical curve at higher percentiles, which can simply be seen in the context of lower sampling size. Figure 2 depicts the L-moment diagrams corresponding to the 12-month SPEI magnitude series. The plots show high dispersion considering the different percentile thresholds. Nevertheless, at low percentiles, the points do not approximate to the theoretical curve of GP distribution, but they conversely tend to approximate to the GP curve at percentiles between 60th and 80th. Again, the points exhibited high dispersion at upper percentiles (mostly above the 85th). An inspection of Supplementary Figure S1 to S14 suggests similar patterns for other timescales and for the drought duration and magnitude series obtained using the SPI. Table 1 summarizes the percentage of the POT series that fit well with the GP distribution following Anderson-Darling statistic. As listed, the series of drought magnitude show better fit to GP distribution than those of drought duration, with no considerable differences between SPI and SPEI. In contrast, we noted remarkable differences as a function of drought timescale. We found that a high percentage of the series obtained for low percentiles does not fit to GP distribution. This fitting improves markedly for all drought duration and magnitude series when considering higher percentiles (mostly above the 40th percentile). The only exceptions are found for the duration series obtained at 1-month time scale using both SPI and SPEI, but considering thresholds higher than 80th percentile. The total percentage of these series is almost close to 100%. Overall, although results suggest that high percentiles (e.g. 90th or 95th) are more appropriate to define the series of drought duration and magnitude, our decision was to define the series using a more relaxed threshold (80th percentile). This decision is motivated by the notion that L-moment statistics show high dispersion at the most upper percentiles. Furthermore, it is difficult to secure enough sampling size considering these upper percentiles. Figure 3 shows the number of drought events corresponding to the different percentiles and timescales (i.e. 1-, 3-, 6-, and 12-month). It can be noted that the number of events using the 90th and 95th percentile thresholds is very low for all timescales. This low number of events is statistically insufficient for reliable estimation of L-moment and GP parameters (Table 2). Accordingly, it is reasonable to consider lower percentiles for making better probabilistic estimations. Our results demonstrate that the series of drought duration and magnitude obtained using 80th percentile as a threshold mostly fit to a GP distribution and the majority of the series ($\approx$ 99%) show solutions for GP parameters. Figure 4 depcits the accuracy metrics (i.e. Willmott's D, MAE and Pearson's r coefficient) which compare the maximum observed and modelled drought duration and magnitude at the grid scale. Results are shown for the four timescales and for the SPEI and SPI. It can be noted that the agreement between the maximum observed and modelled values is higher for drought magnitude series than for

drought duration series. However, either for drought magnitude or duration series, this agreement is improved when considering higher percentiles, especially the 80th percentile. These findings are clearly evident for the SPI and SPEI and also for all selected timescales.

We also compared the empirical and modelled cumulative distribution functions (cdfs) using GP distribution and considering the 80th percentile POT series. The comparisons were made at the pixel scale and considering the two drought indices (SPI vs. SPEI) and the different timescales. A representative example is shown in Figure 5 for the grid point located at 40ºN and 3ºW. As illustrated, there is a high agreement between the empirical and modelled cdfs, irrespective of the drought index and the timescale. However, lower agreement is observed for long timescales (6- and 12-month). This can be expected given the low sampling size at long timescales, in comparison to shorter timescales. Overall, the weighted correlations between the empirical and modelled cdfs show high values (> 0.98) in all cases, which is reflected in the general pattern observed across the whole Spain. Figure 6 shows the spatial distribution of the weighted correlations between the empirical and GP distribution modelled cdfs using the 80th percentile POT series. At 1- and 3-month timescales, the correlations are almost close to 1 for the entire Spain, indicating that the selected scheme of the 80th percentile and the GP distribution are appropriate to statistically model drought duration and magnitude in our study domain.The magnitude of correlation decreases at the 6- and 12-month timescales, though being above 0.97 in most areas.

## 3.2 Mapping drought duration and magnitude

Figures 7 and 8 illustrate the spatial distribution of GP parameters calculated for drought duration series obtained using the SPI and SPEI, respectively. The GP parameters show very similar distributions for the SPI and SPEI. However, there are considerable spatial variations in the distribution of these parameters as a function of the drought timescale, with higher values of the location (Xo) and scale (α) parameters for longer time scales. This can be explained by the increase in drought duration at longer time scales. The shape (k) parameter shows similar range values for all time scales. It is difficult to interpret the geographical distribution of shape (k) due to there is large uncertainty involved in estimating this parameter (Rosbjerg et al., 1992). As illustrated in Supplementary Figure S15 and S16, all parameters show similar spatial patterns for the drought magnitude series.

We mapped drought probability for the drought duration and magnitude series using the parameter maps and Eq. (3). Figure 9 shows the estimated drought duration (in weeks) obtained from the 1-, 3-, 6- and 12-month SPEI series for a period of 50 and 100 years. Results suggest important spatial differences among drought timescales. For example, at the 1-month timescale, the maximum duration is found in central areas of Spain, with more than 40 weeks of consecutive negative SPEI values. The same spatial pattern can also be seen at the 3-month timescale, as central and southern Spain experience longer duration. In northern Spain, the estimated maximum drought duration is almost half than that in central Spain. Nevertheless, the spatial patterns of drought probability differ markedly at the time scales of 6- and 12-months, with the maximum duration found in southeastern and southwestern regions and parts of northern and northeastern Spain. The spatial patterns found at the 12-months timescale closely resemble those observed at 6-months timescale, suggesting a maximum drought

duration (>180 weeks) in a period of 50 years over some regions in the southwest and along the eastern Mediterranean coast. On the other hans, considering the maximum drought duration in a period of 100 years, drought events are expected to extend further, especially in southern Spain. Figure 10 reveals that drought probability maps obtained using the SPI are similar to those obtained using the SPEI, albeit with some spatial differences that can mainly be linked to drought timescale.

Figure 11 summarizes the relationship between the maximum drought duration of the SPEI and SPI, considering 1-, 3-, 6- and 12-month timescale and periods of 50 and 100 years. For drought duration, the agreement between the SPI and SPEI is stronger considering long timescales. For timescales between 1 and 6 months, the SPEI tends to record higher quantile estimates than the SPI. Nevertheless, at 12-month timescale, the differences in the quantile estimates between the two indices are clearly minimized. For drought magnitude, the quantile estimates show more consistent spatial patterns for the two

indices, as compared to those identified for drought duration series (Supplementary Figure S17 to S19).

## 4 Discussion and conclusions

We developed high-resolution drought probability maps for Spain using two widely-recognized drought indices that are comparable spatially and temporally: the Standardized Precipitation Index (SPI) and the Standardized Precipitation Evapotranspiration Index (SPEI). Albeit with their similar conceptual background, they differ in their input variables

necessary for calculation. In specific, while the SPI accounts only for precipitation data (McKee et al., 1993), the SPEI considers the atmospheric evaporative demand in its calculation (Vicente-Serrano et al., 2010). In this study, we computed these two drought indices at different timescales (1-, 3-, 6- and 12-month). The aim was to assess whether there are noticeable spatial differences in the obtained drought hazard probabilities, as a function of the selected index and/or timescale.

We assessed the suitability of the GP distribution to model drought duration and magnitude events. Results demonstrate that drought magnitude and duration series mostly fit well with a GP distribution: a finding that was confirmed in earlier drought assessment investigations in many regions worldwide (e.g. Chen et al., 2011; Serra et al., 2016; Vicente-Serrano and Beguería-Portugués, 2003; Zamani et al., 2015). In this study, our decision was motivated by the need to make balance between the goodness of the fit to the GP distribution on one hand and the selection of a representative threshold to obtain

the POT series on the other hand. Our exploratory analysis suggests the use of the 80th percentile as a threshold. This threshold makes a good balance between the two requirements for the SPI and SPEI and for all timescales.

In earlier hydrologic and climatic hazards investigations, a regionalization approach was applied to estimate the probability distribution, L-moment statistics and the distribution parameters (e.g. Durrans and Tomic, 1996; Serra et al., 2016; She et al., 2014). As opposed to these studies, our preference was given to analyse hazard probability locally. Specifically, in our

calculation of the L-moment statistics and the distribution parameters, we considered each gridded cell as an independent series. While regionalization is advantageous in terms of the spatial homogeneity and the reduction of the parameter uncertainty (Hosking and Wallis, 1997), characterization of drought conditions in our study domain reveals noticeable

spatial differences in reponse to drought time scale. This is clearly evident for probabilities of both drought duration and magnitude. Regionalization is usually based on the variables used for calculating drought indices (i.e. precipitation or difference between precipitation and atmospheric evaporative demand) (Ghosh and Srinivasan, 2016; Habibi et al., 2018; Santos et al., 2011; Yuan et al., 2013; Zhang et al., 2015). Importantly, this study stresses that this kind of regionalization

might not be useful when drought hazard differs strongly as a function of drought timescale. Previous studies indicated that the spatial patterns of drought may strongly differ as a function of drought timescale, especially with the different temporal influence of local/regional precipitation events on drought index values (e.g. Vicente-Serrano 2006). This is confirmed in our study for the whole Spain, where the spatial patterns of GP distribution and the maps of hazard probability strongly vary as a function of the drought timescale. Again, this stresses the difficulty of applying regionalization approaches to obtain maps of

drought probability. This difficulty is also enhanced by our findings on the differences in the drought probability in response to the selected drought index. All together makes this kind of regionalization a challenging task. A possible solution could be establishing different regionalization schemes based on the different series of drought indices and timescales. However, this seems to be practically disadvantageous, making the use of probability estimations by end-users more confused (e.g. stakeholders, decision makers and local communities). Also, with the spatial coherence and the observed gradients of GP

parameters, a direct calculation of hazard probabilities locally is highly recommended, particularly in regions with strong spatial and temporal climatic variability like Spain. Overall, taken all these limitations into consideration, it is recommended to avoid employing regionalization approaches to determine drought hazard probabilities when different drought indices and timescales are used.

Assessing the different spatial patterns of drought probabilities as a function of timescales has strong implications for

drought impact assessment and drought mitigation. It is well-established that different hydrological, agricultural and environmental systems respond differently to drought timescales (Pasho et al., 2012; Peña-Gallardo et al., 2018; Vicente-Serrano, 2013). As such, for more effective assessment and monitoring of drought hazard, drought timescales must be linked with specific drought impacts. This is basically because although drought probability may differ as a function of drought timescale, the impacts of drought hazard can vary considerably from one region to another in response to different

environmental and socioeconomic factors. Correspondingly, the degree of vulnerability can vary according to drought timescale and region. For example, albeit with the high probability of occurrence of an extreme drought event at a certain timescale in a particular region, drought risk may be small if the sensitivity to drought timescale is low. This confirms that it is essential to obtain drought hazard probability maps at different timescales, given that the real hazard would be definitely derived from drought timescale that triggers impacts in a given space and sector.

Recently, there is a great debate on the influence of climate change processes on drought severity (Dai, 2013; Sheffield et al., 2012; Trenberth et al., 2014). This debate is mainly motivated by the role of warming processes and the atmospheric evaporative demand (AED) in drought severity. Numerous studies have shown a noticeable increase in the AED across the Mediterranean region, which could enhance the severity of drought events in comparison to those events caused mainly by precipitation deficit (Stagge et al., 2017; Vicente-Serrano et al., 2014). Here, we indicated that, mainly at timescales from 1-

to 6-month, SPEI duration and magnitude values are higher than those of the SPI, suggesting that increased AED due to warming processes may have certain role in increasing drought duration and magnitude hazard probabilities in Spain. This indicates that when a drought occurs as a consequence of a precipitation deficit, high values of the AED may increase the magnitude and duration of drought events. However, this pattern was not observed with long drought timescales (i.e. 12

5    month), which showed small differences between the SPI and SPEI drought duration and magnitude quantile maps. This could be explained by the strong seasonality that characterises the climate of Spain, considering that the 12-month timescale summarizes the entire annual climate conditions. As indicated by Vicente-Serrano et al. (2014), the role of increased AED (mostly recorded during summer months) would be diminished in comparison to the role of precipitation. In contrast, the role of the AED would be more highlighted at shorter timescales that record stronger seasonal variability.

Assessing drought hazard probability by means of joint probabilities of drought duration and magnitude has been applied in more depth by means of the use of copulas (e.g. Ganguli and Reddy, 2012; Liu et al., 2011; Zhang et al., 2015). Nevertheless, given the nature of the drought indices, the time series exhibit strong temporal autocorrelation and accordingly the duration and magnitude of particular drought events can show high agreement. Here, we found a strong correlation between the magnitude and the duration of drought events for the selected drought indices and timescales. This indicates

that—as expected—the total magnitude of an event is proportional to drought duration, exhibiting similar spatial patterns of drought hazard probability either for drought duration or magnitude series. Therefore, although copulas could give some additional information for particular events, we still believe that an accurate evaluation of drought hazard probability in Spain using a univariate approach is more advantageous.

Given the strong spatial differences in the drought hazard probability over our study domain, the maps obtained in this study

can be useful to improve the management of different sectors, including agriculture, water resources management, urban water supply, tourism, and environmental management. The spatial quantile probabilities developed in this study, combined with those estimated for the 50 and 100 years, are fully accessible for the research community and end-users via the web repository of the Spanish National Research Council (CSIC) at https://digital.csic.es/.

**Author contribution**

All the authors contributed equally to the manuscript.

**Competing interests**

The authors declare that they have no conflict of interest.

**Acknowledgements**

This work was supported by the research projects CGL2014-52135-C03-01 and PCIN-2015-220 financed by the Spanish

Commission of Science and Technology and FEDER, 1560/2015: Herramientas de monitorización de la vegetación mediante modelización ecohidrológica en parques continentales financed by the Red de Parques Nacionales, IMDROFLOOD financed by the Water Works 2014 co-funded call of the European Commission and INDECIS, which is part of ERA4CS, an ERA-NET initiated by JPI Climate, and funded by MINECO with co-funding by the European Union (Grant 690462). Marina Peña-Gallardo was granted by the Spanish Ministry of Economy and Competitiveness, Miquel

Tomas-Burguera was supported by a doctoral grant by the Spanish Ministry of Education, Culture and Sport and Ahmed El Kenawy was supported by a postdoctoral Juan de la Cierva contract.

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

**Table 1 Percentage of the peaks-over-threshold drought duration and magnitude series that fit well with the Generalized Pareto distribution following Anderson-Darling statistic. Results are summarized for different percentiles and timescales using SPEI and SPI.**

| SPEI | Magnitude | | | | Duration | | | |
|------|---------|---------|---------|----------|---------|---------|---------|----------|
|      | 1-month | 3-month | 6-month | 12-month | 1-month | 3-month | 6-month | 12-month |
| 00th | 98.5 | 42.8 | 51.8 | 68.9 | 0.0 | 12.8 | 51.7 | 81.6 |
| 10th | 100.0 | 91.1 | 90.4 | 91.5 | 3.8 | 91.0 | 98.3 | 99.2 |
| 20th | 100.0 | 99.8 | 99.1 | 98.4 | 3.8 | 94.8 | 99.1 | 99.6 |
| 30th | 100.0 | 100.0 | 100.0 | 99.9 | 5.6 | 99.5 | 99.9 | 99.9 |
| 40th | 100.0 | 100.0 | 100.0 | 100.0 | 8.9 | 100.0 | 100.0 | 100.0 |
| 50th | 100.0 | 100.0 | 100.0 | 100.0 | 37.0 | 100.0 | 100.0 | 100.0 |
| 60th | 100.0 | 100.0 | 100.0 | 100.0 | 57.9 | 100.0 | 100.0 | 100.0 |
| 70th | 100.0 | 100.0 | 100.0 | 100.0 | 84.3 | 100.0 | 100.0 | 100.0 |
| 80th | 100.0 | 100.0 | 100.0 | 100.0 | 98.6 | 100.0 | 100.0 | 100.0 |
| 90th | 100.0 | 100.0 | 100.0 | 100.0 | 98.8 | 100.0 | 100.0 | 100.0 |
| 95th | 100.0 | 100.0 | 100.0 | 96.9 | 98.6 | 100.0 | 99.9 | 98.5 |
| **SPI** | 1-month | 3-month | 6-month | 12-month | 1-month | 3-month | 6-month | 12-month |
| 00th | 85.8 | 27.9 | 41.3 | 70.9 | 0.0 | 6.4 | 39.2 | 81.5 |
| 10th | 99.3 | 79.9 | 80.8 | 88.9 | 0.1 | 84.4 | 96.8 | 99.1 |
| 20th | 100.0 | 99.0 | 97.3 | 97.3 | 0.1 | 89.8 | 98.2 | 99.5 |
| 30th | 100.0 | 100.0 | 99.9 | 99.7 | 1.4 | 98.0 | 99.8 | 99.9 |
| 40th | 100.0 | 100.0 | 100.0 | 100.0 | 5.0 | 99.8 | 100.0 | 100.0 |
| 50th | 100.0 | 100.0 | 100.0 | 100.0 | 20.8 | 100.0 | 100.0 | 100.0 |
| 60th | 100.0 | 100.0 | 100.0 | 100.0 | 45.2 | 100.0 | 100.0 | 100.0 |
| 70th | 100.0 | 100.0 | 100.0 | 100.0 | 75.7 | 100.0 | 100.0 | 100.0 |
| 80th | 100.0 | 100.0 | 100.0 | 100.0 | 94.4 | 100.0 | 100.0 | 100.0 |
| 90th | 100.0 | 100.0 | 100.0 | 100.0 | 98.6 | 100.0 | 100.0 | 99.9 |
| 95th | 100.0 | 100.0 | 100.0 | 98.2 | 97.1 | 99.9 | 99.9 | 98.8 |

**Table 2 Percentage of cases in which solution for the L-moment and the Generalized Pareto distribution parameters is found for the peaks over threshold drought duration/magnitude series at different percentiles from 1-, 3-, 6-, and 12-month SPI and SPEI.**

|  | 1-month SPEI | 3-month SPEI | 6-month SPEI | 12-month SPEI | 1-month SPI | 3-month SPI | 6-month SPI | 12-month SPI |
|---|---|---|---|---|---|---|---|---|
| 00th | 100.0 | 100.0 | 100.0 | 100.0 | 100.0 | 100.0 | 100.0 | 100.0 |
| 10th | 100.0 | 100.0 | 100.0 | 100.0 | 100.0 | 100.0 | 100.0 | 100.0 |
| 20th | 100.0 | 100.0 | 100.0 | 100.0 | 100.0 | 100.0 | 100.0 | 100.0 |
| 30th | 100.0 | 100.0 | 100.0 | 100.0 | 100.0 | 100.0 | 100.0 | 100.0 |
| 40th | 100.0 | 100.0 | 100.0 | 100.0 | 100.0 | 100.0 | 100.0 | 100.0 |
| 50th | 100.0 | 100.0 | 100.0 | 100.0 | 100.0 | 100.0 | 100.0 | 100.0 |
| 60th | 100.0 | 100.0 | 100.0 | 100.0 | 100.0 | 100.0 | 100.0 | 99.9 |
| 70th | 100.0 | 100.0 | 100.0 | 99.6 | 100.0 | 100.0 | 99.9 | 99.2 |
| 80th | 100.0 | 100.0 | 99.7 | 97.4 | 100.0 | 99.9 | 99.3 | 96.8 |
| 90th | 99.7 | 98.5 | 96.8 | 79.8 | 99.5 | 97.7 | 96.6 | 84.9 |
| 95th | 98.7 | 86.7 | 75.9 | 52.7 | 96.8 | 91.1 | 85.0 | 52.8 |

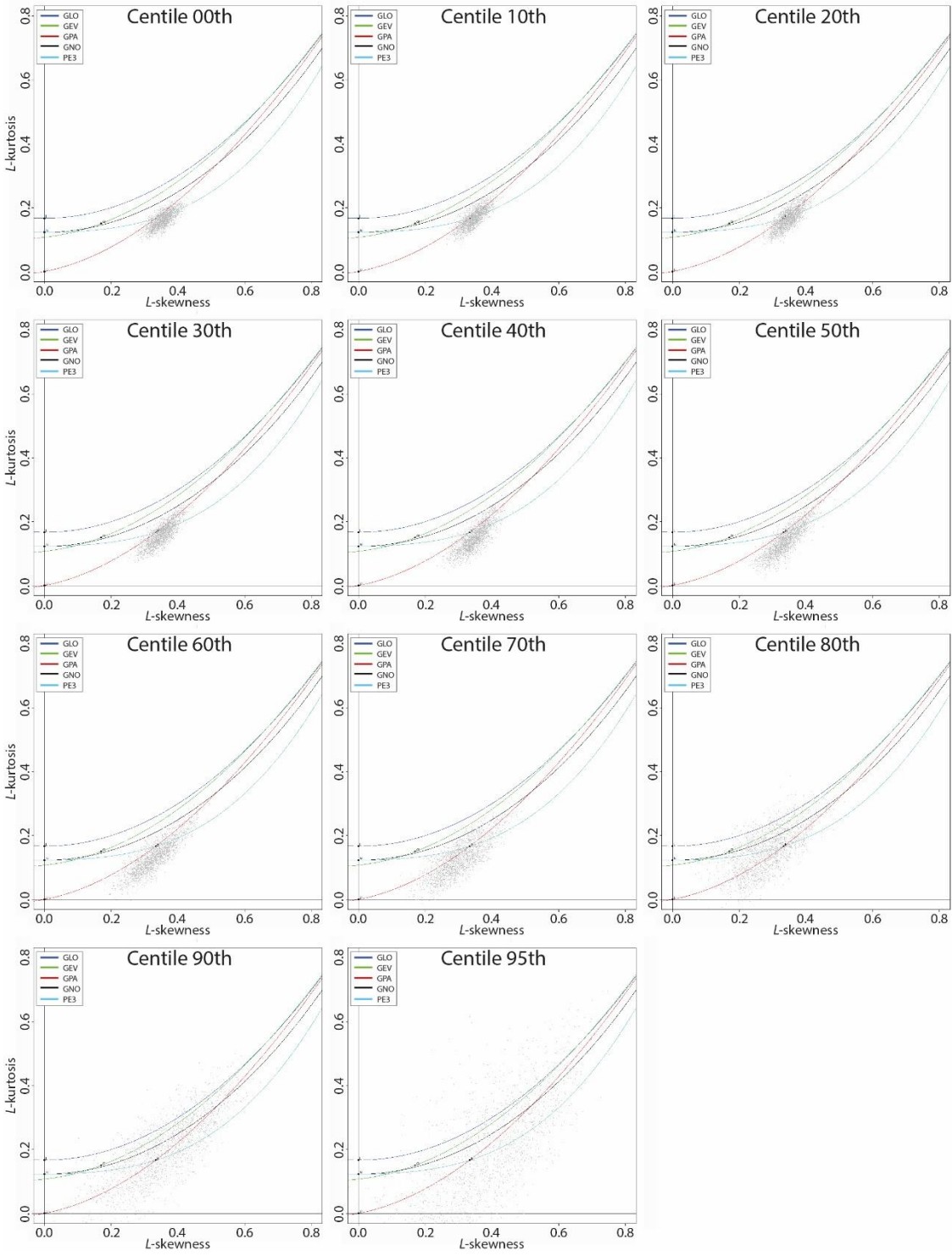

**Figure 1: L-moment diagrams for the peak-over-threshold series obtained from the 1-month SPEI duration series.**

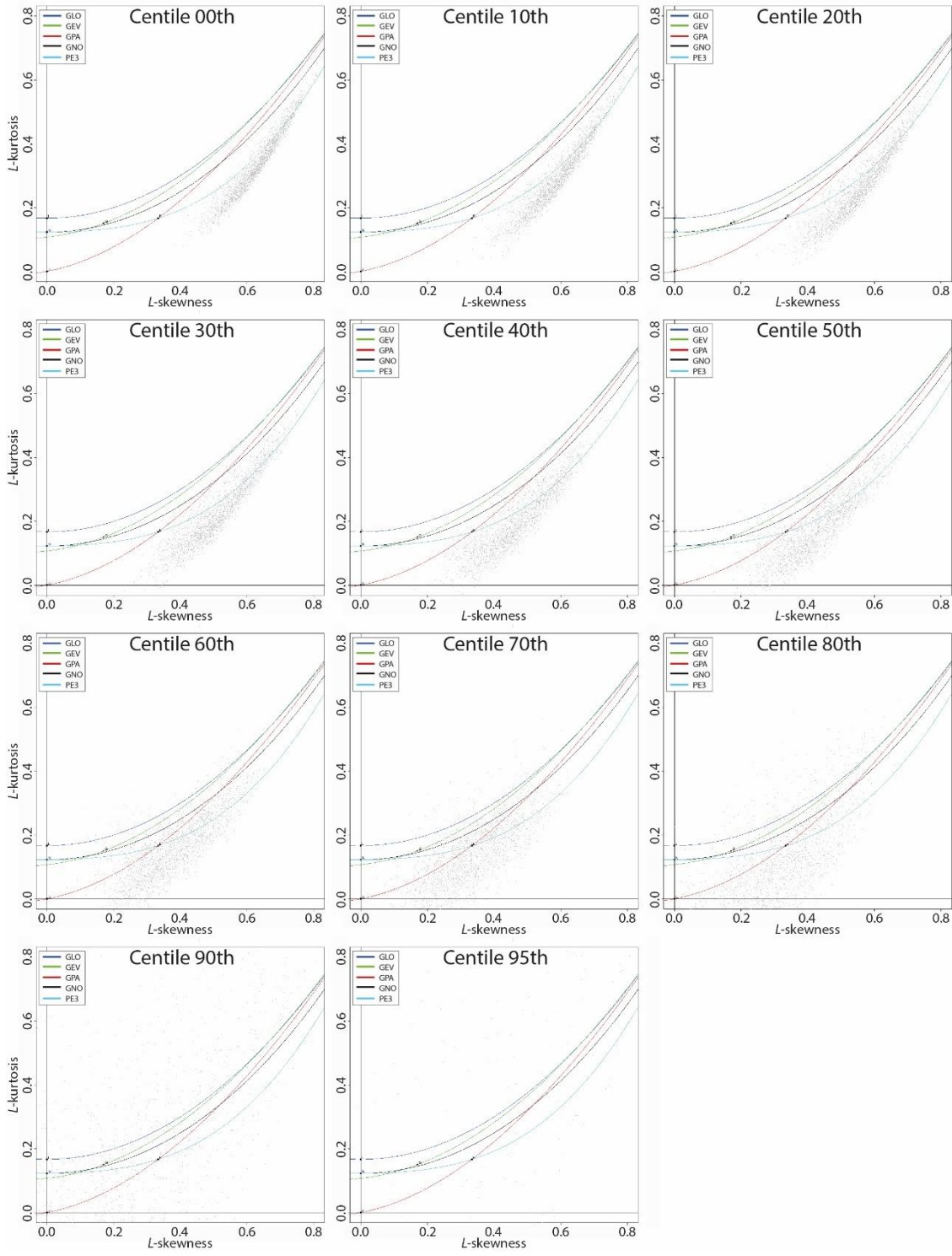

**Figure 2: L-moment diagrams for the peak-over-threshold series obtained from the 12-month SPEI magnitude series.**

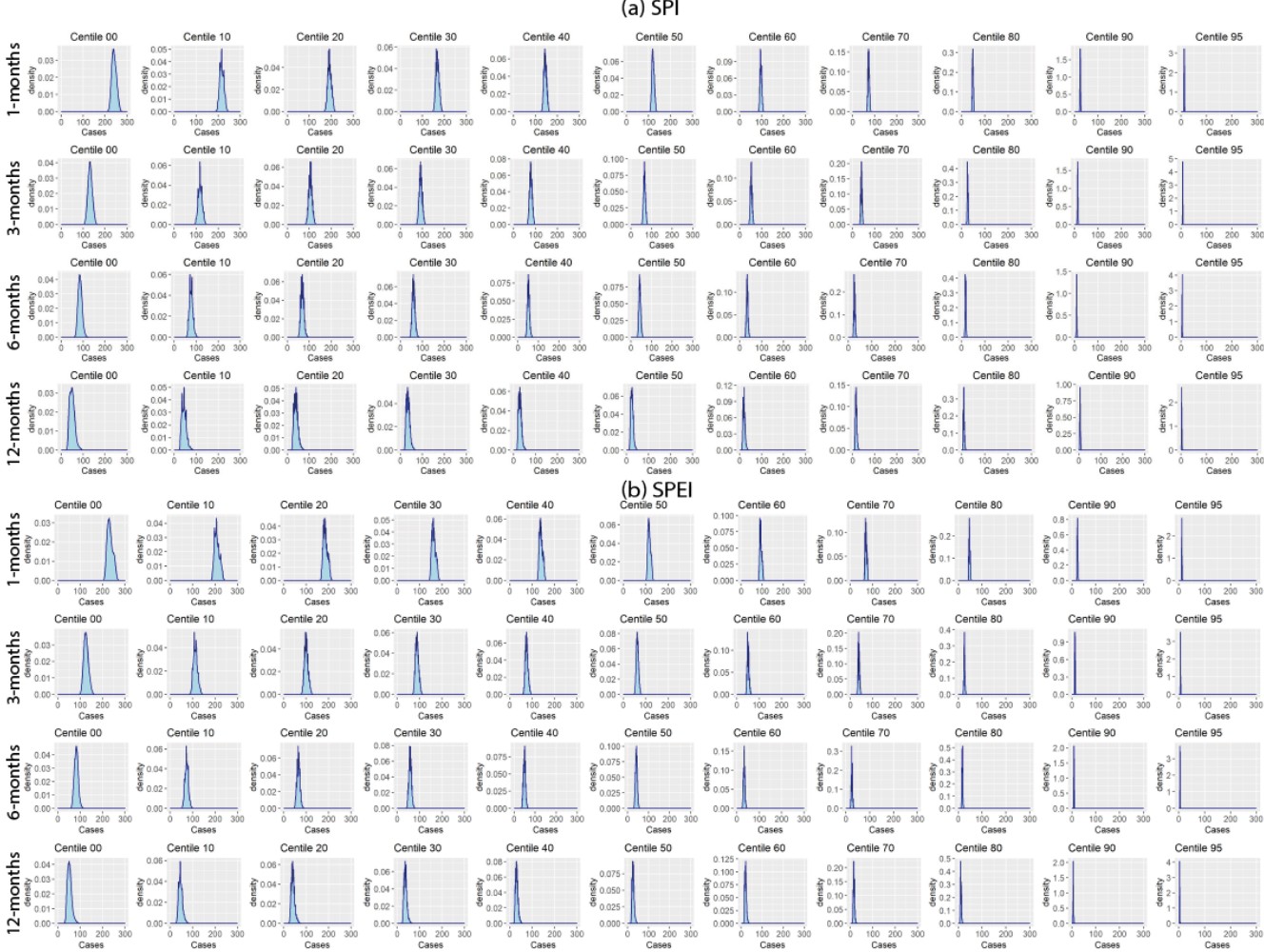

5    Figure 3: Probability density diagrams showing the number of cases corresponding to the peaks over threshold drought
     duration/magnitude series at different percentiles and different timescales (1-, 3-, 6-, and 12-month) using (a) SPI and (b) SPEI.

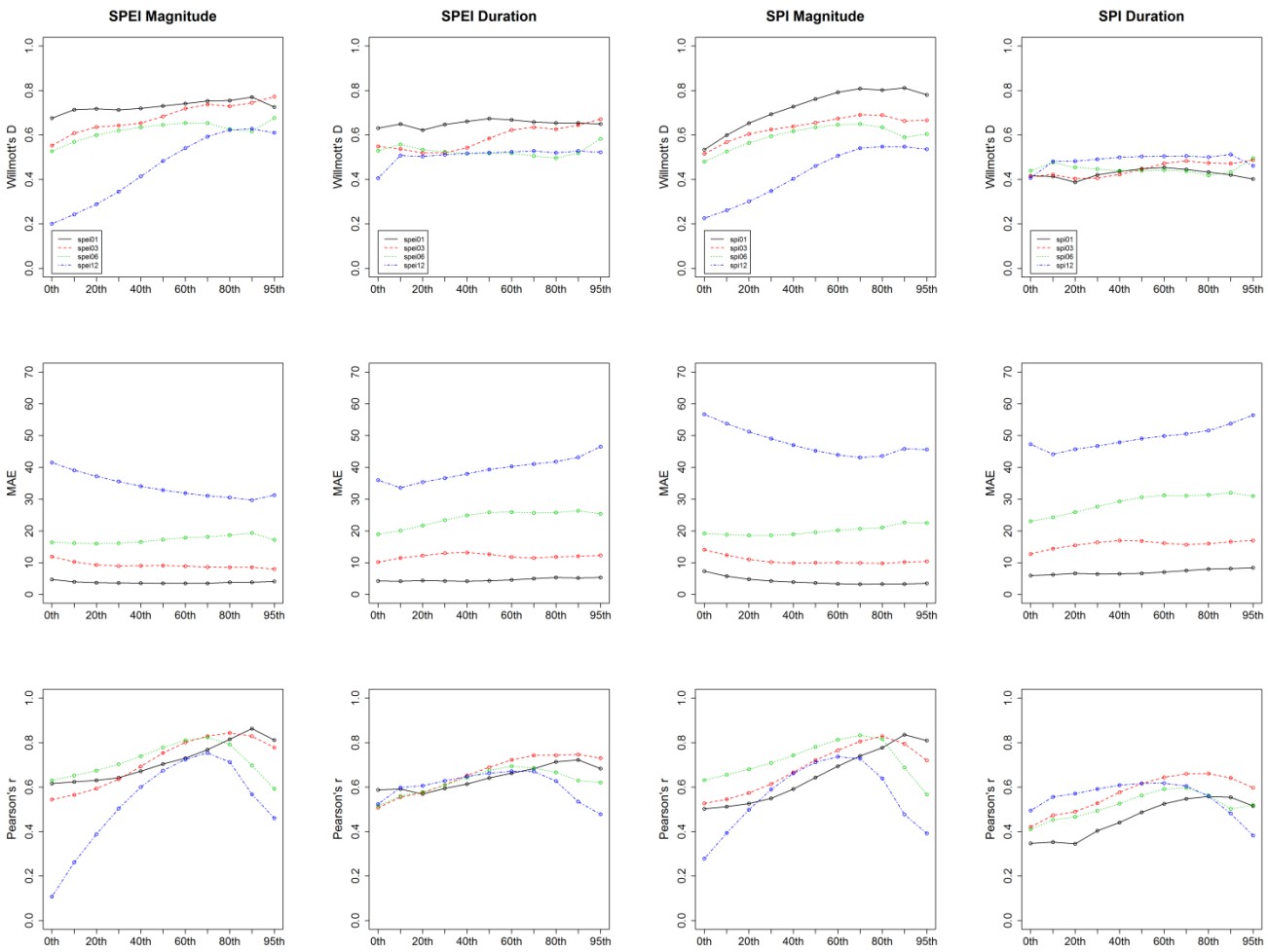

**Figure 4: Willmott's D, mean absolute error (MAE) and Pearson's r, summarized as a function of the different percentiles used to obtain the peaks-over-threshold series. All accuracy metrics were computed based on comparing the maximum observed and modelled 1-, 3-, 6- and 12-month SPI and SPEI drought duration and magnitude between 1961 and 2014. The modelled data were computed using the Generalized Pareto distribution.**

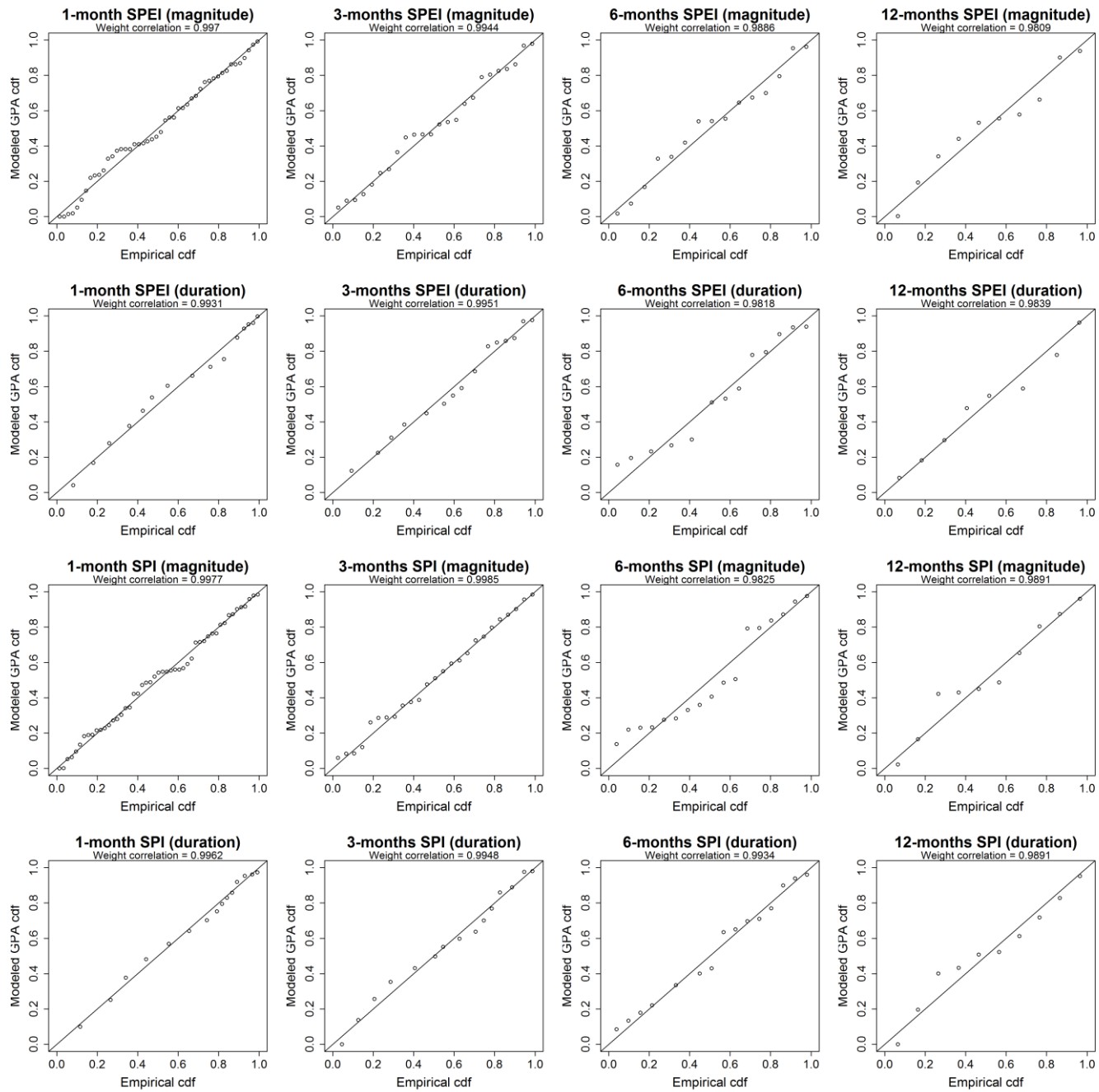

**Figure 5: Example of probability-probability (P-P) plots for the series of 1-, 3-, 6- and 12-month SPEI and SPI drought duration and drought magnitude obtained by means of the 80th percentile used as a threshold to derive the peak-over-threshold series.**

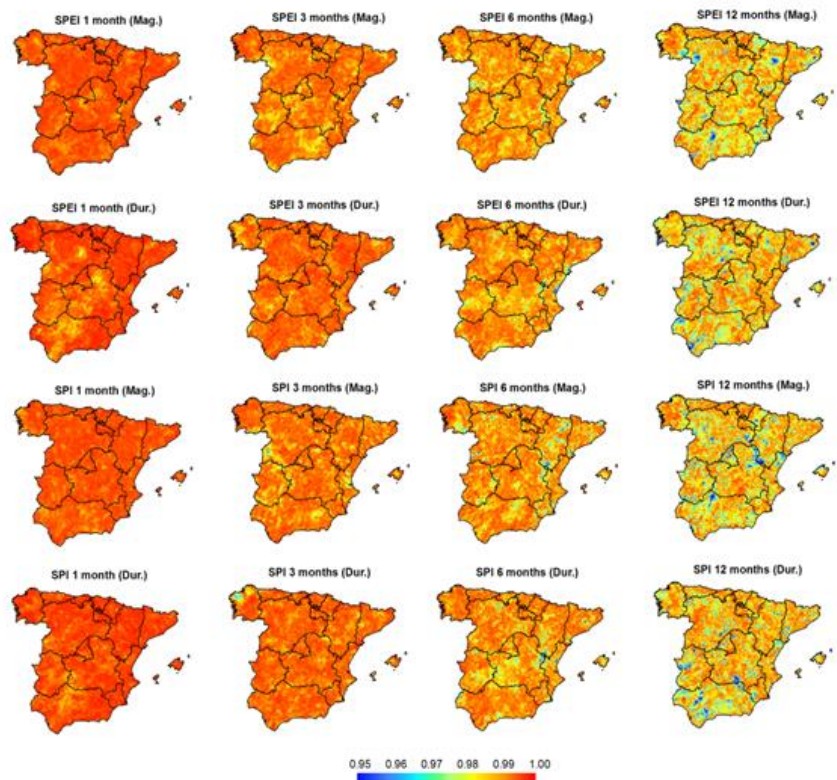

**Figure 6: Spatial distribution of the weight correlation coefficients from probability-probability (P-P) plots from the series of 1-, 3- , 6- and 12-month SPEI and SPI drought duration and magnitude series obtained considering the 80th percentile as a threshold for the peak-over-threshold series.**

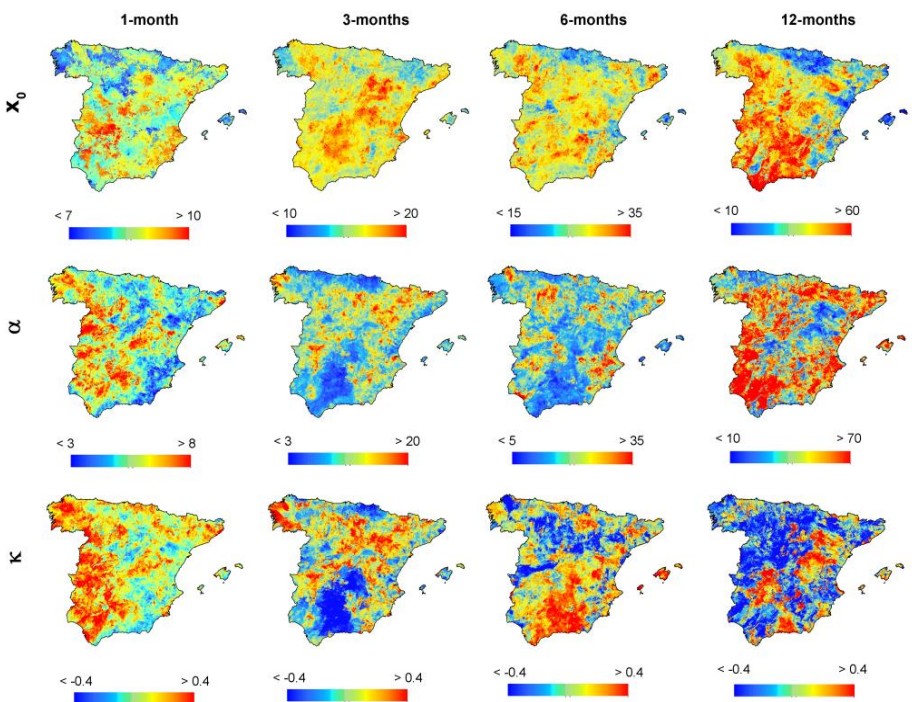

**Figure 7: Spatial distribution of the parameters of the GP distribution for the SPI duration series.**

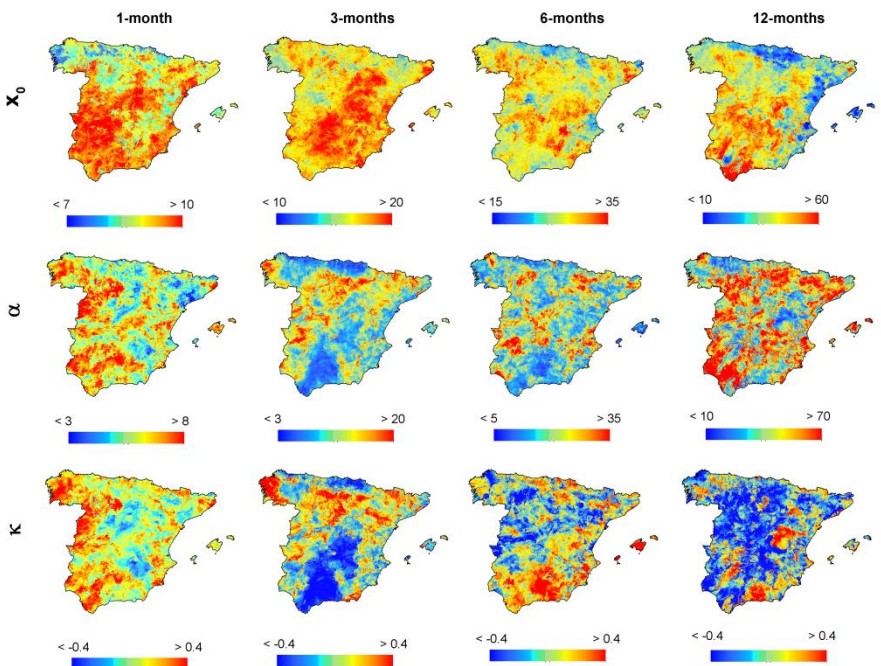

**Figure 8: Spatial distribution of the parameters of the GP distribution for the SPEI duration series.**

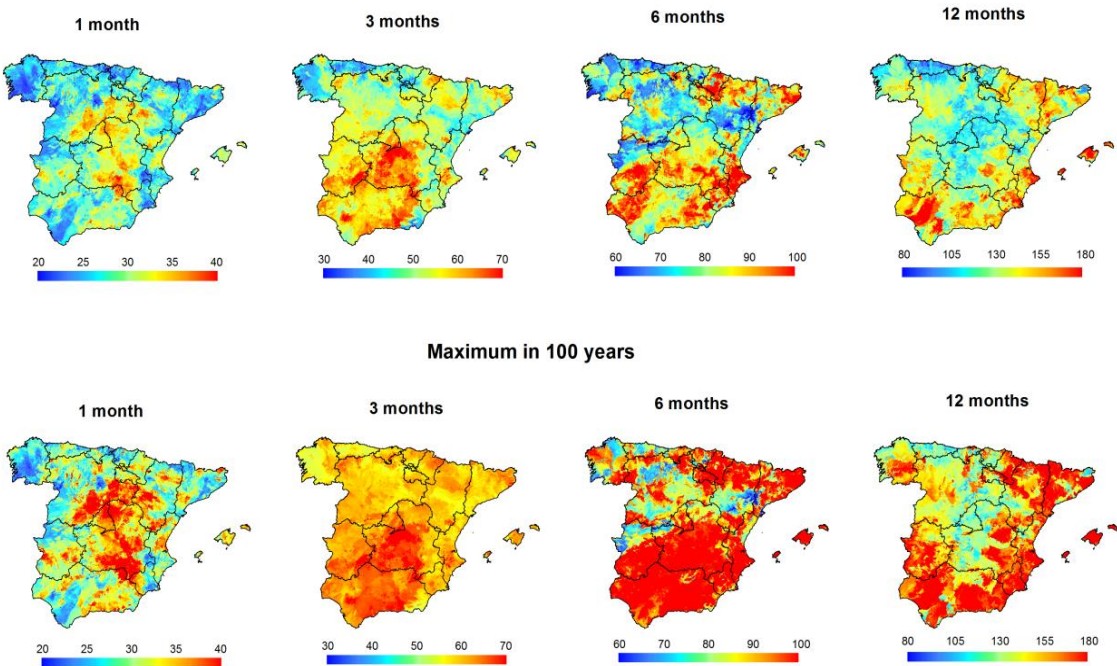

**Figure 9: Spatial distribution of the maximum drought duration (in weeks) from the 1-, 3-, 6- and 12-month SPEI series in a period of 50 and 100 years.**

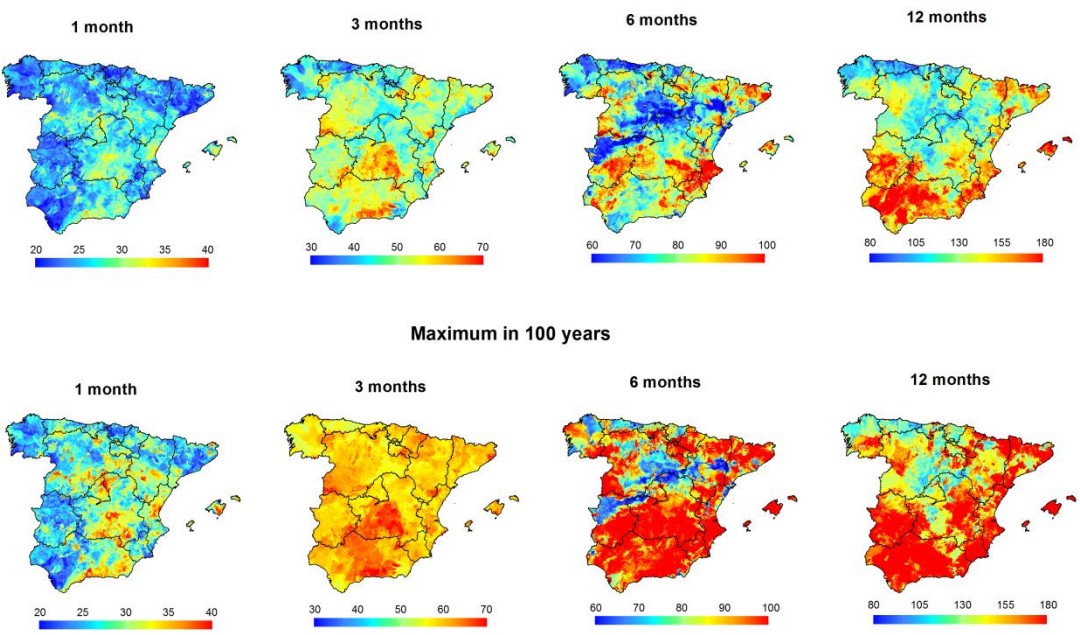

**Figure 10: Spatial distribution of the maximum drought duration (in weeks) from the 1-, 3-, 6- and 12-month SPI series in a period of 50 and 100 years.**

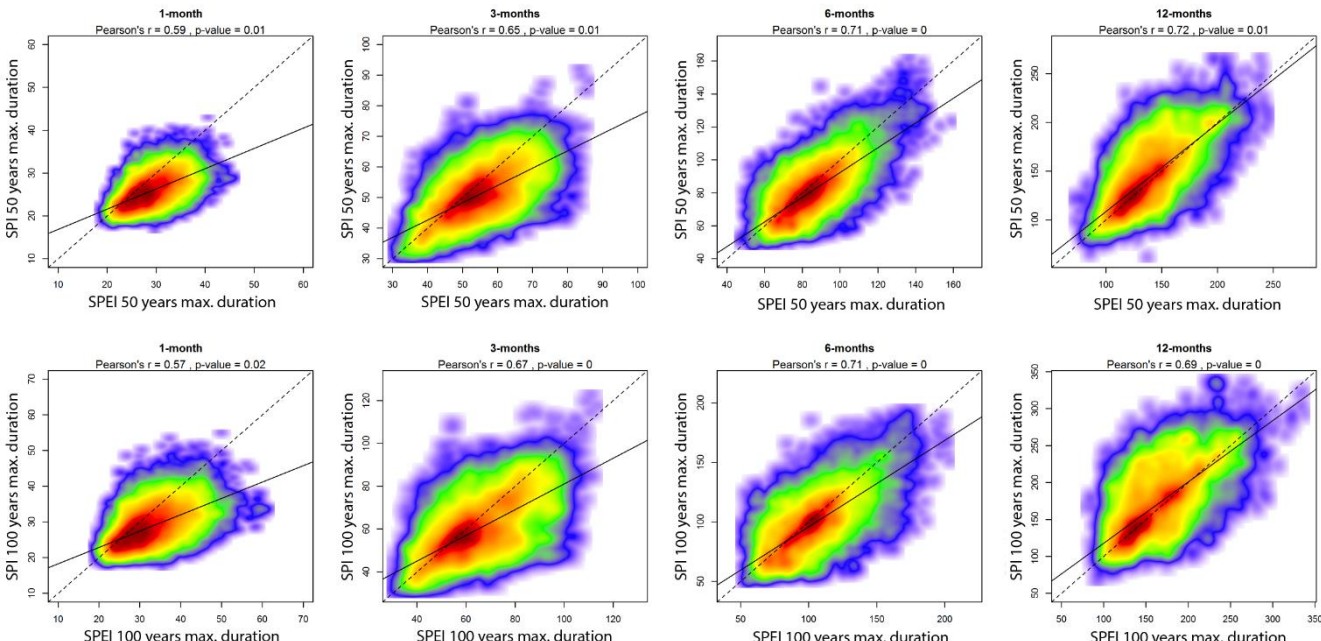

**Figure 11: Scatterplots showing the relationship between the maximum drought duration (in weeks) expected in a period of 50 and 100 years considering 1-, 3-, 6- and 12- SPEI and the SPI. Colors represent the density of points, with red denoting the highest density. Given the large sample used, the significance of the Pearson's r coefficients was estimated by means of a Montecarlo approach using $10^3$ random samples, with each sample containing 30 cases.**