# Peer review of "High-spatial resolution probability maps of drought duration and magnitude across Spain"

_Natural Hazards and Earth System Sciences, 2018_

## Referee Comment (RC1) · Anonymous Referee #1 · 30 Dec 2018

The topic of this paper is very actual and important taking into account recent extreme climate events and what we will face in near future. The authors have presented probabilities of extreme drought events in terms of drought duration and magnitude. They compared two the most used drought indices (SPI and SPEI) at 4 time scales and high spatial resolution by applying extreme value theory. In order to increase readability and relevance of the paper some points need to be addressed.

Some parts of the text are hard to understand so I would recommend English check before final publication. Please use term "percentile" instead of "centile". Overuse of the adverb "Nevertheless".

[Figure]

The Abstract should be rewritten as well as the Section "Data and methods" since it is not clear in every step what is done on what data and it would not be understandable to wider audience.

Through the selection of the appropriate threshold you observed percentiles from 0th to 95th. Why haven't you take values greater than e.g. 40th or 50th percentile since you are studying extremes? Then you would have less figures and they would be more visible. Figures are hardly visible (Supplement ones even less), especially if they are printed in black and white. Lines and dots should be thicker.

L10 – "-for the first time-" should be removed. Make unique terminology – drought severity > drought magnitude. It is not the same. L14 "…implying that drought event is attained only when the index values are lower than zero."It is not "implied". It is according to index definition. Or I didn't understand the sentence well. L15 "drought severity and magnitude series" > "drought duration and magnitude series" L16 "evaluating different three-parametric distributions" – in the text you are mentioning only one distribution, have you tested some more? L29 What are the "large legislation practices"?

L10-L14 If the stated studies "developed drought-related probability maps for Spain" how they "did not account for the different drought hazard probability"? I understand the point you want to make, but the text should be reformulated. L20 What do you mean by "normalizing data of climatic variables for common periods"? > Suggestion "climate data standardization over standard climatological periods" L23 There are various drought indices including ones that do not account for the climatology of the location that is observed, as you explained in part L14-20. You should specify the group of drought indices you are talking about. L25-26 Sentence "Taken together…." should be reformulated.

L6 "Unfortunately, this aspect has receives less attention in the literature." Is this your opinion? Can you somehow confirm this statement? If not, it should be omitted. L19 Datasets cover the whole Spain. How many points are in the grid you considered? L19-20What is the source of the meteorological parameters? L21 Have you calculated the indices or are the datasets for the indices downloaded from the website you provided? L23 What is "normalization of the climatic balance"? L24 Which method have you used for PET calculation? L28 Chosen timescales include agricultural drought as well, and agriculture is very important sector. It should be included.

It should be stated here that the indices include wet conditions as well, and the indices values below 0 signify drought condition. You mentioned it in the Abstract but not in the text. For which time period have you calculated distribution parameters for indices calculation?

L1-10 I think this part is unnecessary and brings a bit of confusion in the paper. According to the indices definition, drought event is when the indices values are below 0. For each drought event you calculated its magnitude and duration and then extracted extreme ones using POT. There was no need to introduce run theory. L13 I am curious how you did the integration. I don't have access to Dracup paper, so if you could be more specific on this. L15 "drought duration was calculated for the consecutive weeks..." > "drought duration was calculated as number of consecutive weeks..." L19 Stationarity is referred to series of drought magnitude and duration, right? L26-30 In the abstract you are mentioning "evaluating different three-parametric distributions". Have you tested some other distributions beside GP? On the Figure 1 are presented L moment diagrams for different distributions, but in the text there are no explanations regarding them.

L3-4 Repetition of the Page 4 L26-30 L6-L13 This part should be rewritten and be one paragraph. The first sentence "Hosking (1990)...." Is hanging and it is connected to the sentence in L2 of the previous paragraph but I cannot see its point in the whole text? Sentences from L8 till L12 refer to the same thing: you plotted L-moment diagrams and applied Anderson-Darling test to obtain and test POT series, fitting to GP distribution, for different x0 thresholds; am I right? L17 "t years" or "T years"? L19 What is "original sample" in this case? L14-L21 There should be one paragraph referring to maximum duration and magnitude.

L1 Formula is not explained, i.e. elements of formula are not defined. L9 "Nevertheless... 1-month timescale can be different considering other drought timescales." It can be, but is it in your study? L4-11 Is there any specific reason why you have chosen to present SPEI1 for duration and SPEI12 for magnitude? L13 There are too many supplementary figures regarding this part. Can they be reduced and just described in the text. There are no significant differences between them (I think, the dots on the Figures are barely visible on some panels...is there any for 95th percentile?). L15 Have you done Anderson-Darling statistic for other distributions? What are the other distributions that you have tested? L20-21 "Notably...." This sentence refers to 1-month scale or general? L26 Could you please say something more on Fig 3? Does Fig 3 unify duration and magnitude? Scales on panels are different (even for the same percentile) so they are hardly comparable. L28 What did you do in the cases when you could not calculate distribution parameters? Is this 99% referring to percent of series for both indices, magnitude and duration, all time scales and all grid points? L29 "A comparison of the observations and estimations..." where we can see this?

L4 "Similar results ...." Does it mean that previous sentences are related to other two metrics? L6 "Again ... at the pixel scale..." - what else was compared on pixel scale?

As I understood, previous paragraph was on maximum duration/magnitude over the whole grid. L7 Have you plotted cdfs for every grid point and then chosen the representative example? Why this grid point? Could you put some mark on this point on Fig 6. Do you have any idea why there is smaller correlation in some locations, are there any specific geographical characteristics that influence the results (e.g. in NW for SPI3 duration, Fig.6)? L18-L21 What the differences mean? In general, what these figures are showing, what you can conclude from them? It is notable that $\kappa$ changes the sign between 3 and 6 month timescale for all cases (SPI/SPEI and duration/magnitude), do you have idea why? L22 "We mapped drought probability…. using the parameter maps and Eq.3" – How did you do this, using some GIS software? L31 "southeast" or "southwest"?

Supplementary Fig 17, 18 - What are "SPI/SPEI units"? L31 You mention "climatic balance" again. Climate balance is based on the balance between various components of climate system. I doubt you are referring to them all, so this terminology is not correct.

L6 "This difficulty is also enhanced by our findings on the spatial differences in the drought probability in response to the selected drought index." But in Page 7/8 you say "Drought probability maps using the SPI show spatial patterns similar to those observed by means of the SPEI". So, are there significant spatial differences in the drought probability comparing two indices? L21 "As such, the degree of vulnerability can vary according to drought timescale" – I would add "drought timescale and region."

Technical corrections

Please make unique way of representation in tables and figures: order of SPI and SPEI as well as "duration" and "magnitude".

L26 Add "e.g." in the brackets since you stated only two articles published in 2018. L30 "practices to drought events" – should it be "practices during drought events"?

L10 "several works" > "several studies" L17-18 "wet conditions" > "moisture conditions" "km2" > "km2"

L3 "varying" > "various" L11 "...drought event as that event with a period..." > "... drought event as period..."

L2 "(Hosking, 1990)" > "Hosking (1990)" L4 "world regions" > "regions of the world"

L9 "low agreement" > "lower agreement" (because in preceding sentence you said "very good agreement" for all; not to be contradicted) L23 "predicted" > "estimated" L28 "12-month" > "12-months" L31 ">180 months" > ">180 weeks"

L10 "Standardized precipitation Index" > "Standardized Precipitation Index" L19 "was made to make balance" > "was to make balance"

L3 & L7 Check the references (names of the authors)

In Table 1, "SPEI", the "I" went to the second row. Figure 3 Both lower and upper panels have the same name "SPEI" Figure 6 There is number "40" on panels SPI 1, 3, 6 for duration. Figure 11 Axis labels "duracion" > "duration" Supplementary Fig 19 - Figure

caption is for duration.

---

## Referee Comment (RC2) · Anonymous Referee #2 · 2 Jan 2019

The paper presents a methodology to characterize drought duration and intensity over Spain using two climatic indices: SPI and SPEI. The work uses a gridded dataset of SPI and SPEI values calculated weekly at high spatial resolution over Spain. From this dataset and using SPEI and SPI at four different time scales, the authors obtain a peak-over-threshold empirical series of drought duration and magnitude on which they fit a Pareto distribution. The fitted probability distribution is then used to produce maps of the maximum drought duration and magnitude of different time scales. The work differs from previous drought characterization efforts in the high spatial and temporal resolution at which the study is conducted, and the use of a GP distribution to capture the probability distribution of of extreme anomalies, which is critical for correct drought

characterization.

I found the study valuable from the methodological point of view and from the insight it provides on drought patterns. The paper provides important methodological guidance on the most adequate probability distribution to characterize exceedance thresholds through a thorough analysis of different candidate probability distributions that could represent the POT series. It also shows that the spatial patterns of drought can be very different when droughts are characterized at different time scales. I do not have major methodological concerns, however, the paper is written such that some methodological and conceptual aspects are not clear. Part of the problem is that the paper needs to be heavily edited for language and style. Also, the authors need to pay attention to details. For instance, some of the symbols used in the equations are not defined in the main text or the symbols used in the text and the equation are different (e.g. x0, w_j). The labels of Figure 1, 2 and 3 cannot be read and their general quality need to be improved. There are many awkwardly written sentences throughout the paper that are distracting and detract from the quality of the study. The paper, as currently written, is not ready for publication.

Section 2.2. discusses the arbitrary nature of selecting thresholds in the indices to define drought, and how these thresholds are different for different activities or processes impacted by drought. Then in Page 4 line 6 says that the studies uses an 'arbitrary' threshold of zero and define drought as an event with an index below zero. Isn't it the standard way of applying these indices to define drought? In that case, zero represents the long term average climatology and therefore it could be argued it is not an arbitrary threshold. The way the paragraph is written makes me doubt whether I am actually interpreting this correctly. I suggest that paragraph is edited to be more specific or clear about what the authors actually mean. Also, a few additional details in the methodology, such as how were the climatic inputs used to produce the indices gridded, may help interpret the results.

I have a few additional questions: why does the paper use the word centile instead of

percentile? Does it have a specific meaning, like the percentile from the empirical cdf? Page 4 line 21: does the 0th percentile actually exist? Does it refer to the minimum value in the record? Page 7 line 9-10: I am not sure you should expect that low model-observation agreement is caused by the lower sampling size at long time scales. Why would that be? Goodness of fit and robustness are different things.

———————————————————

---

## Author Comment (AC1) · 14 Feb 2019

Anonymous Referee #1 The topic of this paper is very actual and important taking into account recent extreme climate events and what we will face in near future. The authors have presented probabilities of extreme drought events in terms of drought duration and magnitude. They compared two the most used drought indices (SPI and SPEI) at 4 time scales and high spatial resolution by applying extreme value theory. In order to increase readability and relevance of the paper some points need to be addressed. Many thanks; we appreciate your positive comment and recommendations.

Some parts of the text are hard to understand so I would recommend English check

before final publication. Please use term "percentile" instead of "centile". Overuse of the adverb "Nevertheless". The text has been polished by a professional English speaker. We have replaced "centile" by "percentile" in the whole manuscript. We have also reduced the use of "Nevertheless" in the text.

The Abstract should be rewritten as well as the Section "Data and methods" since it is not clear in every step what is done on what data and it would not be understandable to wider audience. All these sections have been revised carefully and we believe that the flow of the text and the readiability of these sections are highly improved.

Through the selection of the appropriate threshold you observed percentiles from 0th to 95th. Why haven't you take values greater than e.g. 40th or 50th percentile since you are studying extremes? Then you would have less figures and they would be more visible. Figures are hardly visible (Supplement ones even less), especially if they are printed in black and white. Lines and dots should be thicker. We completely understand the concerns of the reviewer. However, recalling that this study provides the first comprehensive assessment of drought probability in the whole Spain, we are keen to evaluate all the percentiles to understand better the studied variables. In accordance with the reviewer's comment, we have improved the resolution of the figures to make their readiability much easier.

Page 1 L10 – "-for the first time-" should be removed. Make unique terminology – drought severity > drought magnitude. It is not the same. Amended.

L14 ". . .implying that drought event is attained only when the index values are lower than zero." It is not "implied". It is according to index definition. Or I didn't understand the sentence well. We have rewritten the sentence. However, it is noteworthy indicating that there are no global standard criteria to define drought events. McKee et al. (1993) adopted a threshold of SPI lower than -1 or -0.8 to select drought events. In this study, a drought index value below zero was defined. This threshold allows for detecting all drought events, regardless of duration and severity. Our motivation of this selection is

explained in the new version of the text, as: "We used an arbitrarily-defined threshold (equal to zero) to define drought events. Although this threshold allows for inclusion of less severe drought events, it can secure a sufficient sampling size to conduct the probabilistic analysis. Importantly, the retention of drought events in this manner will not bias the obtained results, given that high values of the series will be retained following the peak-over-threshold approach".

L15 "drought severity and magnitude series" > "drought duration and magnitude series" We have changed.

L16 "evaluating different three-parametric distributions" – in the text you are mentioning only one distribution, have you tested some more? We have deleted this sentence from the abstract to avoid ambiguity. We represent the generalized logistic, generalized extreme value, generalized Pareto, generalized normal and Pearson type III, in Figures 1,2 and supplementary Figures 1-14. However, these figures show that Generalized Pareto is clearly the distribution with the best fit. The outperformance of GP distribution has already been confirmed in many earlier studies. In specific, many studies demonstrate that the probability distribution of a POT series with random occurrence times fits well with GP distribution (see for example, Hosking et al., 1987; Pham et al., 2014; Wang, 1991). This is why we have not analyzed the findings of other statistical distributions. Hosking, J. R. M. and Wallis, J. R..: Parameter and quantile estimation for the generalized pareto distribution, Technometrics, 29(3), 339–349, doi:10.1080/00401706.1987.10488243, 1987 Pham, H. X., Asaad, Y. and Melville, B.: Statistical properties of partial duration series: Case study of North Island, New Zealand, J. Hydrol. Eng., 19(4), 807–815, doi:10.1061/(ASCE)HE.1943-5584.0000841, 2014. Wang, Q. J.: The POT model described by the generalized Pareto distribution with Poisson arrival rate, J. Hydrol., 129(1–4), 263–280, doi:10.1016/0022-1694(91)90054-L, 1991.

L29 What are the "large legislation practices"? The most important legislation practice in Spain is the Special Drought Plans. There is one Plan per hydrological basin. In

2018, the new plans were approved, which substitute the earlier plans of 2007. We have included these Plans as examples in the text . "In addition to the different monitoring systems for hydrological drought conditions (Maia and Vicente-Serrano, 2017), there are national legislation practices that aim to improve drought adaptation strategies and practices, such as Special Droughts Plans"

Page 2 L10-L14 If the stated studies "developed drought-related probability maps for Spain" how they "did not account for the different drought hazard probability"? I understand the point you want to make, but the text should be reformulated. We have rewritten the text, as follows: "In Spain, several studies have developed dry spells probability maps (e.g. Lana et al., 2006; Martin-Vide and Gomez, 1999; Pérez-Sánchez and Senent-Aparicio, 2018). However, given that the probability of occurrence of dry spells is higher in arid regions than in humid regions, these studies did not account for the different drought hazard probability across Spain. It is well-recognized that the frequency and duration of dry spells are closely driven by the climatology of the studied area. As such, it can be expected that a simple map of climate aridity in Spain can show similar spatial patterns to those of dry spell probability". Lana, X., Martínez, M. D., Burgueño, A., Serra, C., Martín-Vide, J. and Gómez, L.: Distributions of long dry spells in the Iberian Peninsula, years 1951-1990, Int. J. Climatol., 26(14), 1999–2021, doi:10.1002/joc.1354, 2006. Martin-Vide, J. and Gomez, L.: Regionalization of peninsular Spain based on the length of dry spells, Int. J. Climatol., 19(5), 537–555, 1999. Pérez-Sánchez, J. and Senent-Aparicio, J.: Analysis of meteorological droughts and dry spells in semiarid regions: a comparative analysis of probability distribution functions in the Segura Basin (SE Spain), Theor. Appl. Climatol., 133(3–4), 1061–1074, doi:10.1007/s00704-017-2239-x, 2018

L20 What do you mean by "normalizing data of climatic variables for common periods"? > Suggestion "climate data standardization over standard climatological periods" We have followed the suggestion, as: "This highlights the importance of data standardization to make drought characteristics (e.g. duration, intensity, severity) comparable

among regions with different climatic conditions".

L23 There are various drought indices including ones that do not account for the climatology of the location that is observed, as you explained in part L14-20. You should specify the group of drought indices you are talking about. Amended. We have include some examples "e.g. Standardized Precipitation Index, Standardized Precipitation Evapotranspiration Index, Palmer Drought Severity Index, Self-calibrated Palmer Drought Severity Index"

L25-26 Sentence "Taken together. . .." should be reformulated. We have reformulated as: "Overall, based on these drought indices, it is possible to map the probability of occurrence of drought duration and magnitude at a detailed spatial resolution."

Page 3 L6 "Unfortunately, this aspect has receives less attention in the literature." Is this your opinion? Can you somehow confirm this statement? If not, it should be omitted. We have omitted this statement.

L19 Datasets cover the whole Spain. How many points are in the grid you considered? We have 1115 * 834 pixels, in which 412178 pixels have data, while other pixels correspond to the Mediterranean Sea and Atlantic Ocean. We have included this information in the text, as: ". . .developed a high-resolution spatial (1.21 km2) and temporal (weekly) drought dataset for Spain (412178 pixels)."

L19- 20 What is the source of the meteorological parameters? The raw data were provided by the National Spanish Meteorological Services (AEMET). After a careful check of the quality and homogeneity, the raw data of each climatic variable was interpolated to create a gridded dataset for the whole Spain. This has been clarified in the text, as: "Based on gridded datasets of maximum and minimum air temperatures (1304 observatories), precipitation (2269 observatories), wind speed (82 observatories), relative humidity (179 observatories) and sunshine duration (112 observatories), Vicente-Serrano et al. (2017) developed. . ."

L21 Have you calculated the indices or are the datasets for the indices downloaded from the website you provided? Vicente-Serrano et al. (2017) developed the indices. We have rewritten the text to clarify this point. Vicente-Serrano, S. M., Tomas-Burguera, M., Beguería, S., Reig, F., Latorre, B., Peña-Gallardo, M., Luna, M. Y., Morata, A. and González-Hidalgo, J. C.: A High Resolution Dataset of Drought Indices for Spain, Data, 2(3), 22, doi:10.3390/data2030022, 2017.

L23 What is "normalization of the climatic balance"? We have considered this comment, as: "SPEI is based on normalization of the difference between precipitation and atmospheric evaporative demand"

L24 Which method have you used for PET calculation? The method to compute ET0 was the reference FAO-56 Penman-Monteith (Allen et al., 1998). R.G. Allen, L.S. Pereira, D. Raes Crop evapotranspiration—guidelines for computing crop water requirements FAO Irrigation and drainage paper 56 (1998)

L28 Chosen timescales include agricultural drought as well, and agriculture is very important sector. It should be included. It should be stated here that the indices include wet conditions as well, and the indices values below 0 signify drought condition. You mentioned it in the Abstract but not in the text. For which time period have you calculated distribution parameters for indices calculation? Agricultural droughts are very important in Spain. The damages to agricultural are generally related to 3-, 6- month time scales (Pacoa et al., 2017; Peña-Gallardo et al., 2019). We believe that the readers of NHESS are aware of the the SPI and SPEI. As such, we believe that it is not necessary to indicate that each index has positive values that reveal wet conditions. Our decision to define drought events using a SPEI/SPI threshold of zero is clearly explained: "We used an arbitrarily-defined threshold (equal to zero) to define drought events. Although this threshold allows for inclusion of less severe drought events, it can secure a sufficient sampling size to conduct the probabilistic analysis. Importantly, the retention of drought events in this manner will not bias the obtained results, given that high values of the series will be retained following the peak-over-threshold approach".

The distribution parameters for the indices are calculated for the whole period 1961-2014. Pascoa P, Gouveia CM, Russo A, Trigo RM (2017) The role of drought on wheat yield interannual variability in the Iberian Peninsula from 1929 to 2014. Int J Biometeorol 61:439–451. Peña-Gallardo, M., Vicente-Serrano, S. M., Domínguez-Castro, F., and Beguería, S.: The impact of drought on the productivity of two rainfed crops in Spain, Nat. Hazards Earth Syst. Sci. Discuss., https://doi.org/10.5194/nhess-2019-1, in review, 2019

"Page 4 L1-10 I think this part is unnecessary and brings a bit of confusion in the paper. According to the indices definition, drought event is when the indices values are below 0. For each drought event you calculated its magnitude and duration and then extracted extreme ones using POT. There was no need to introduce run theory. We have deleted the reference to run theory. But we consider appropriate the justification of the election of 0 as a threshold. Probably some readers were familiarized with other thresholds as -0.8 or -1 to define drought periods

L13 I am curious how you did the integration. I don't have access to Dracup paper, so if you could be more specific on this. L15 "drought duration was calculated for the consecutive weeks. . ." > "drought duration was calculated as number of consecutive weeks. . ." It is a simple integration of the values below 0 as you can see in figure 1 from López-Moreno et al. (2009). López-Moreno J.I, Vicente-Serrano S.M., Beguería S., García-Ruiz J.M., Portela M.M., Almeida A.B. Dam effects on drought magnitude and duration in a transboundary basin: The lower River Tagus, Spain and Portugal. Water resources research 45, W02405.

L19 Stationarity is referred to series of drought magnitude and duration, right? Definitiely.

L26-30 In the abstract you are mentioning "evaluating different three-parametric distributions". Have you tested some other distributions beside GP? On the Figure 1 are presented L moment diagrams for different distributions, but in the text there are no

explanations regarding them. We have deleted this sentence from the abstract pleas se the comment to the P1. L16.

Page 5 L3-4 Repetition of the Page 4 L26-30 Thank, we have delete this sentence to avoid repetitions.

L6-L13 This part should be rewritten and be one paragraph. The first sentence "Hosking (1990). . .." Is hanging and it is connected to the sentence in L2 of the previous paragraph but I cannot see its point in the whole text? The whole section 2.3 has been rewritten and the readiability has improved.

Sentences from L8 till L12 refer to the same thing: you plotted L-moment diagrams and applied Anderson-Darling test to obtain and test POT series, fitting to GP distribution, for different x0 thresholds; am I right? Yes you are.

L17 "t years" or "T years"? Amended.

L19 What is "original sample" in this case? It refers to drought duration and magnitude over the whole study period (1961-2014).

L14-L21 There should be one paragraph referring to maximum duration and magnitude. We have included a reference to the maximum duration and magnitude.

Page 6 L1 Formula is not explained, i.e. elements of formula are not defined. We have included a description of all symbols included in this equation.

L9 "Nevertheless... 1-month timescale can be different considering other drought timescales." It can be, but is it in your study? No, it is not. We have omitted this sentence because it is already stated in L 12. Figs S1-S14 suggest similar patterns for other timescales as well as for the drought duration and magnitude series obtained using the SPI.

L4-11 Is there any specific reason why you have chosen to present SPEI1 for duration and SPEI12 for magnitude? We attempted only to give an example of short and

long timescale. Results corresponding to all timescales are already presented in the supplementary material.

L13 There are too many supplementary figures regarding this part. Can they be reduced and just described in the text. There are no significant differences between them (I think, the dots on the Figures are barely visible on some panels. . .is there any for 95th percentile?). We prefer to provide all the information in the supplementary material. As NHESS is an online journal, all readers can easily access the supplementary material. Moreover, the limitation of the journal to the supplementary material is 50MB and we are very far from this limit.

L15 Have you done Anderson-Darling statistic for other distributions? What are the other distributions that you have tested? We have not considered this statistic for other distributions. As illustrated in Figs. 1,2, and S1-S14, our results confirm that the best fit is recorded with GP distribution. This finding concurs well with what earlier studies suggest (Hosking et al., 1987; Pham et al., 2014; Wang, 1991). Hosking, J. R. M. and Wallis, J. R..: Parameter and quantile estimation for the generalized pareto distribution, Technometrics, 29(3), 339–349, doi:10.1080/00401706.1987.10488243, 1987 Pham, H. X., Asaad, Y. and Melville, B.: Statistical properties of partial duration series: Case study of North Island, New Zealand, J. Hydrol. Eng., 19(4), 807–815, doi:10.1061/(ASCE)HE.1943-5584.0000841, 2014. Wang, Q. J.: The POT model described by the generalized Pareto distribution with Poisson arrival rate, J. Hydrol., 129(1–4), 263–280, doi:10.1016/0022-1694(91)90054-L, 1991.

L20-21 "Notably. . .." This sentence refers to 1-month scale or general? To the 1-month time scale, we have revised this statement to make it clear, as : "The only exceptions are found for the duration series obtained at 1-month time scale using both SPI and SPEI, but considering thresholds higher than 80th percentile. The total percentage of these series is almost close to 100%".

L26 Could you please say something more on Fig 3? Does Fig 3 unify duration and

magnitude? Scales on panels are different (even for the same percentile) so they are hardly comparable. Figure 3 is the number of cases, so it is the same for duration and magnitude. We have included a sentence explaining the main result of the figure. "Figure 3 shows the number of drought events corresponding to the different percentiles and timescales (i.e. 1-, 3-, 6-, and 12-month). It can be noted that the number of events using the 90th and 95th percentile thresholds is very low for all timescales. This low number of events is statistically insufficient for reliable estimation of L-moment and GP parameters (Table 2)".

L28 What did you do in the cases when you could not calculate distribution parameters? Is this 99% referring to percent of series for both indices, magnitude and duration, all time scales and all grid points? Yes, in less than 1% of all series, we were not able to calculate the parameters. In such cases, we excluded these pixels (series) from subsequent analyses.

L29 "A comparison of the observations and estimations. . ." where we can see this? This can be seen in Figure 4. However, we have deleted this sentence to avoid any misunderstanding. Page 7

L4 "Similar results . . .." Does it mean that previous sentences are related to other two metrics? We have deleted this statement.

L6 "Again . . . at the pixel scale. . ." - what else was compared on pixel scale? As I understood, previous paragraph was on maximum duration/magnitude over the whole grid. There is an error. The "Again" is not correct. We have corrected as: "The comparisons were made at the pixel scale. . ."

L7 Have you plotted cdfs for every grid point and then chosen the representative example? Why this grid point? Could you put some mark on this point on Fig 6. Do you have any idea why there is smaller correlation in some locations, are there any specific geographical characteristics that influence the results (e.g. in NW for SPI3 duration, Fig.6)? Indeed, it is extremely difficult to plot cdfs for a total number of 412.178 pixels.

Rather, we randomly selected one pixel. We clarified the coordinates of this grid (L21). It is quite difficult to mark this point in Figure 6, as it will mask hundreds of pixels. We cannot see a clear geographical pattern representing locations with small correlation. As such, it is difficult to provide an explanation of this pattern. Probably, it can be linked to the generation of the gridded SPI and SPEI values.

L18-L21 What the differences mean? In general, what these figures are showing, what you can conclude from them? It is notable that $\kappa$ changes the sign between 3 and 6 month timescale for all cases (SPI/SPEI and duration/magnitude), do you have idea why? We have included a paragraph describing and interpreting both figures, as follows. "Figures 7 and 8 illustrate the spatial distribution of GP parameters calculated for drought duration series obtained using the SPI and SPEI, respectively. The GP parameters show very similar distributions for the SPI and SPEI. However, there are considerable spatial variations in the distribution of these parameters as a function of the drought timescale, with higher values of the location (Xo) and scale ($\alpha$) parameters for longer time scales. This can be explained by the increase in drought duration at longer time scales. The shape (k) parameter shows similar range values for all time scales. It is difficult to interpret the geographical distribution of shape (k) due to there is large uncertainty involved in estimating this parameter (Rosbjerg et al., 1992). As illustrated in Supplementary Figure S15 and S16, all parameters show similar spatial patterns for the drought magnitude series". Rosbjerg, D., Madsen, H., Rasmussen, P.F.: Prediction in partialduration series with generalized Pareto-distributed exceedances, Water Resources Research, 28(11): 3001 – 3010, 1992

L22 "We mapped drought probability. . .. using the parameter maps and Eq.3" – How did you do this, using some GIS software? R was used to produce this figure. It is a publicly free software (https://cran.r-project.org/bin/windows/base/).

L31 "southeast" or "southwest"? This is an unexpected error that we corrected in the new version of the manuscript.

[Figure]

Interactive
comment

Page 8 Supplementary Fig 17, 18 - What are "SPI/SPEI units"? Both indices are normalized using a probability distribution function, so that values of SPI or SPEI are actually seen as standard deviations from the median. We have changed to z-units.

L31 You mention "climatic balance" again. Climate balance is based on the balance between various components of climate system. I doubt you are referring to them all, so this terminology is not correct. We have rewritten as: "i.e. precipitation or difference between precipitation and atmospheric evaporative demand".

Page 9 L6 "This difficulty is also enhanced by our findings on the spatial differences in the drought probability in response to the selected drought index." But in Page 7/8 you say "Drought probability maps using the SPI show spatial patterns similar to those observed by means of the SPEI". So, are there significant spatial differences in the drought probability comparing two indices? We have reformulated the sentence of page 7/8 to stress the differences between the SPI and SPEI. "The aim was to assess whether there are noticeable spatial differences in the obtained drought hazard probabilities, as a function of the selected index and/or timescale".

L21 "As such, the degree of vulnerability can vary according to drought timescale" – I would add "drought timescale and region." We have included.

Technical corrections Please make unique way of representation in tables and figures: order of SPI and SPEI as well as "duration" and "magnitude". Page 1 L26 Add "e.g." in the brackets since you stated only two articles published in 2018. L30 "practices to drought events" – should it be "practices during drought events"? Page 2 L10 "several works" > "several studies" L17-18 "wet conditions" > "moisture conditions" "km2" > "km2" Page 4 L3 "varying" > "various" L11 ". . .drought event as that event with a period. . ." > ". . . drought event as period. . ." Page 5 L2 "(Hosking, 1990)" > "Hosking (1990)" L4 "world regions" > "regions of the world" Page 7 L9 "low agreement" > "lower agreement" (because in preceding sentence you said "very good agreement" for all; not to be contradicted) L23 "predicted" > "estimated" L28 "12-month" > "12-months"

L31 ">180 months" > ">180 weeks" Page 8 L10 "Standardized precipitation Index" > "Standardized Precipitation Index" L19 "was made to make balance" > "was to make balance" Page 12 L3 & L7 Check the references (names of the authors) In Table 1, "SPEI", the "I" went to the second row.

Many thanks for all these technical corrections. We have modified the text following your recommendations.

Figure 3 Both lower and upper panels have the same name "SPEI" We appreciate this comment. The figure was wrong, we have corrected in the new version of the manuscript.

Figure 6 There is number "40" on panels SPI 1, 3, 6 for duration. We have deleted the "40".

Figure 11 Axis labels "duracion" > "duration" Amended.

Supplementary Fig 19 – Figure caption is for duration Amended.
* * *
[Figure]

**Figure 3.** Example of the smoothing procedure employed for standardized precipitation index (SPI) series and calculation of drought magnitude and duration.

**Fig. 1.**

---

## Author Comment (AC2) · 14 Feb 2019

The paper presents a methodology to characterize drought duration and intensity over Spain using two climatic indices: SPI and SPEI. The work uses a gridded dataset of SPI and SPEI values calculated weekly at high spatial resolution over Spain. From this dataset and using SPEI and SPI at four different time scales, the authors obtain a peak-over-threshold empirical series of drought duration and magnitude on which they fit a Pareto distribution. The fitted probability distribution is then used to produce maps of the maximum drought duration and magnitude of different time scales. The work differs

from previous drought characterization efforts in the high spatial and temporal resolution at which the study is conducted, and the use of a GP distribution to capture the probability distribution of extreme anomalies, which is critical for correct drought characterization. I found the study valuable from the methodological point of view and from the insight it provides on drought patterns. The paper provides important methodological guidance on the most adequate probability distribution to characterize exceedance thresholds through a thorough analysis of different candidate probability distributions that could represent the POT series. It also shows that the spatial patterns of drought can be very different when droughts are characterized at different time scales. I do not have major methodological concerns, however, the paper is written such that some methodological and conceptual aspects are not clear.

Many thanks for your positive comment.

Part of the problem is that the paper needs to be heavily edited for language and style. The text has been polished by a professional English speaker to improve the language style and text flow.

Also, the authors need to pay attention to details. For instance, some of the symbols used in the equations are not defined in the main text or the symbols used in the text and the equation are different (e.g. x0, w_j). The symbols corresponding to each equation are described in the new version of the manuscript, and we have systematized the symbols in the text and equations

The labels of Figure 1, 2 and 3 cannot be read and their general quality need to be improved. We have improved the quality of these figures.

There are many awkwardly written sentences throughout the paper that are distracting and detract from the quality of the study. The paper, as currently written, is not ready for publication. The text has been polished by a professional English speaker to improve the language style and text flow.

Section 2.2. discusses the arbitrary nature of selecting thresholds in the indices to define drought, and how these thresholds are different for different activities or processes impacted by drought. Then in Page 4 line 6 says that the studies uses an 'arbitrary' threshold of zero and define drought as an event with an index below zero. Isn't it the standard way of applying these indices to define drought? In that case, zero represents the long term average climatology and therefore it could be argued it is not an arbitrary threshold. The way the paragraph is written makes me doubt whether I am actually interpreting this correctly. I suggest that paragraph is edited to be more specific or clear about what the authors actually mean. Globally, there is no standard definition of drought. Likewise, there are different procedures/methods to define a drought event, with no standard threshold. All negative SPI or SPEI values characterize drought conditions, regardless of the magnitude or severity. Some studies applies a threshold of -0.8 or -1 z-unit to define a drought event. We have clarified the rationale behind our selection of zero unit, as a threshold, as follows. "There are several criteria (thresholds) to identify independent drought events (e.g. Fleig et al., 2006; Lee et al., 1986). These thresholds are generally arbitrary, with no clear objective metrics to relate a certain value of a drought index with specific sectorial impacts. Indeed, this is a challenging task, given the large number of economic sectors and environmental systems impacted by droughts (Pérez and Barreiro-Hurlé, 2009). Furthermore, regions and sectors can respond differently to various drought timescales (Lorenzo-Lacruz et al., 2013; Pasho et al., 2012). In this work, we obtained the series of drought events from the weekly gridded series of SPEI and SPI at four selected time scales (1-, 3-, 6- and 12-months). We used zero threshold to define drought events. Although this threshold allows for inclusion of less severe drought events, it can secure a sufficient sampling size to conduct the probabilistic analysis. Importantly, the retention of drought events in this manner will not bias the obtained results, given that high values of the series will be retained following the peak-over-threshold approach". Fleig, A. K., Tallaksen, L. M., Hisdal, H. and Demuth, S.: A global evaluation of streamflow drought characteristics, Hydrol. Earth Syst. Sci., 10(4), 535–552, doi:10.5194/hess-10-535-2006, 2006

Lee, K. S., Sadeghipour, J. and Dracup, J. A.: An Approach for Frequency Analysis of Multiyear Drought Durations, Water Resour. Res., 22(5), 655–662, doi:10.1029/WR022i005p00655, 1986

Lorenzo-Lacruz, J., MoÅŢan-Tejeda, E., Vicente-Serrano, S. M. and Äźopez-Moreno, J. I.: Streamflow droughts in the Iberian Peninsula between 1945 and 2005: Spatial and temporal patterns, Hydrol. Earth Syst. Sci., 17(1), 119–134, doi:10.5194/hess-17-119-2013, 2013.

Pasho, E., Camarero, J. J. and Vicente-Serrano, S. M.: Climatic impacts and drought control of radial growth and seasonal wood formation in Pinus halepensis, Trees - Struct. Funct., 26(6), 1875–1886, doi:10.1007/s00468-012-0756-x, 2012.

Pérez, L. and Barreiro-Hurlé, J.: Assessing the socio-economic impacts of drought in the Ebro River Basin | Análisis de los efectos socioeconómicos de la sequía en la cuenca del Ebro, Spanish J. Agric. Res., 7(2), 269–280, 2009.

Also, a few additional details in the methodology, such as how were the climatic inputs used to produce the indices gridded, may help interpret the results. We have included a more detailed description of how the gridded indices were produced. "Based on gridded datasets of maximum and minimum air temperatures (1304 observatories), precipitation (2269 observatories), wind speed (82 observatories), relative humidity (179 observatories) and sunshine duration (112 observatories), Vicente-Serrano et al. (2017) developed a high-resolution spatial (1.21 km2) and temporal (weekly) drought dataset for Spain (412178 pixels). This dataset spans the period from 1961 to 2014. Importantly, this drought dataset was developed after a rigorous procedure to check the quality and homogeneity of the input climatic data. The grid of each variable was computed by universal kriging method (Borrough and McDonnell 1998; Pebesma, 2004), using latitude, longitude and elevation of each grid cell as auxiliary variables. The grid layers were validated with a jackknife resampling procedure (Phillips et al., 1992) and difference between the predicted and observed values for

each observatory was low (see Vicente-Serrano et al., 2017 for details). Overall, the gridded climatic data were employed to compute the Standardized Precipitation Index (SPI) (McKee et al., 1993) and the Standardized Precipitation Evapotranspiration Index (SPEI) (Vicente-Serrano et al., 2010) at different timescales ranging from 1- to 48-month (http://monitordesequia.csic.es). While the SPI accounts only for precipitation data, the SPEI is based on normalization of the difference between precipitation and atmospheric evaporative demand (AED). In this study, we employed these two drought indices to assess the possible impacts of the AED on drought hazard probability in Spain. The SPI and SPEI were used at time scales of 1-, 3-, 6- and 12-months for the period 1961-2014".

Borrough, P. A., & McDonnell, R. A. (1998). Principles of Geographical Information Systems. UK, Oxford University Press.

Pebesma, E. J. (2004). Multivariable geostatistics in S: The gstat package. Comput. Geosci., 30, 683–691.

Phillips, D. L., Dolph, J., & Marks, D. (1992). A comparison of geostatistical procedures for spatial analysis of precipitation in mountainous terrain. Agric. Meteorol. 58, 119–141.

Vicente-Serrano, S. M., Tomas-Burguera, M., Beguería, S., Reig, F., Latorre, B., Peña-Gallardo, M., Luna, M. Y., Morata, A. and González-Hidalgo, J. C.: A High Resolution Dataset of Drought Indices for Spain, Data, 2(3), 22, doi:10.3390/data2030022, 2017.

McKee, T. B., Doesken, N. J. and Kleist, J.: The relationship of drought frequency and duration to time scales, Eighth Conf. Appl. Climatol., 179–184, 1993.

Vicente-Serrano, S. M., Beguería, S. and López-Moreno, J. I.: A multiscalar drought index sensitive to global warming: The standardized precipitation evapotranspiration index, J. Clim., 23(7), 1696–1718, doi:10.1175/2009JCLI2909.1, 2010.

I have a few additional questions: why does the paper use the word centile instead of

percentile? Does it have a specific meaning, like the percentile from the empirical cdf? In the whole manuscript, we have replaced "centile" with "percentile".

Page 4 line 21: does the 0th percentile actually exist? Does it refer to the minimum value in the record? Indeed, the 0th percentile considers all the serie.

Page 7 line 9-10: I am not sure you should expect that low model observation agreement is caused by the lower sampling size at long time scales. Why would that be? Goodness of fit and robustness are different things. At longer scales there are less events, so we have lower sampling size. This clearly affects the accuaracy of the esti­amtions. The extreme cases are the pixels that have no solution for the GP parameters, which also increase for longer scales (see Table 2).